# EReLiFM: Evidential Reliability-Aware Residual Flow Meta-Learning for Open-Set Domain Generalization under Noisy Labels

## Abstract

Open-Set Domain Generalization (OSDG) aims to enable deep learning models to recognize unseen categories in new domains, which is crucial for real-world applications. Label noise hinders open-set domain generalization by corrupting source-domain knowledge, making it harder to recognize known classes and reject unseen ones. While existing methods address OSDG under Noisy Labels (OSDG-NL) using hyperbolic prototype-guided meta-learning, they struggle to bridge domain gaps, especially with limited clean labeled data. In this paper, we propose Evidential Reliability-Aware Residual Flow Meta-Learning (EReLiFM). We first introduce an unsupervised two-stage evidential loss clustering method to promote label reliability awareness. Then, we propose a residual flow matching mechanism that models structured domain- and category-conditioned residuals, enabling diverse and uncertainty-aware transfer paths beyond interpolation-based augmentation. During this meta-learning process, the model is optimized such that the update direction on the clean set maximizes the loss decrease on the noisy set, using pseudo labels derived from the most confident predicted class for supervision. Experimental results show that EReLiFM outperforms existing methods on OSDG-NL, achieving state-of-the-art performance. The source code is available at `https://anonymous.4open.science/r/ERELIFM-CBCB`.

## 1 Introduction

Open-Set Domain Generalization (OSDG) tackles both domain and category shifts, requiring models to classify known categories while rejecting unseen ones. It is critical in dynamic applications such as healthcare Li et al. (2020), security Busto et al. (2020), and autonomous driving Guo et al. (2022), where new domains and categories often arise. Recent works employ meta-learning Wang et al. (2023); Shu et al. (2021) to simulate cross-domain tasks during training, improving adaptability to novel environments. Yet, one can never expect the annotation to be $100\%$ correct. Label noise further complicates OSDG by compromising the reliability of knowledge learned from source domains. This challenges existing OSDG approaches as introduced in Peng et al. (2024a). Although label noise has been extensively studied in standard classification tasks, it remains largely unaddressed in OSDG.

Existing techniques, such as relabeling Zhang et al. (2024); Zheng et al. (2020); Li et al. (2024), data pruning Kim et al. (2021); Karim et al. (2022), and loss-based noise-agnostic methods Xu et al. (2024); Yue & Jha (2024) focus on refining training data by correcting mislabeled instances or through selective optimization based on loss values. However, these methods do not address the additional challenge of adapting to unseen domains and distinguishing novel categories, which is essential in OSDG. Peng et al. (2024a) introduced novel benchmarks for the task of OSDG under Noisy Labels (OSDG-NL) based on widely-used PACS Li et al. (2017) and DigitsDG Zhou et al. (2020a) datasets. Related approaches from both the OSDG and noisy label learning fields are evaluated as baselines.

HyProMeta Peng et al. (2024a) serves as the first solution developed specifically targeting OSDG-NL, where hyperbolic prototypes are used to guide meta-learning optimization. Label noise agnostic meta-learning in HyProMeta is achieved by computing hyperbolic category prototypes to separate clean and noisy samples based on hyperbolic distances, correcting noisy labels using nearest prototypes, and augmenting training with a learnable prompt to enhance generalization to unseen categories.

However, prototype-based classification in HyProMeta is limited by sensitivity to noise and feature quality, which results in a negative effect on label noise diagnosis. Due to the limited number of clean samples and limited label-clean/noisy partition capability, HyProMeta suffers from unsatisfactory generalization performance, as less trustworthy a priori can be provided for the label-noise-agnostic meta-learning.

In this work, we propose a new method, *i.e.*, Evidential Reliability-Aware Residual Flow Meta-Learning (EReLiFM). Our method introduces a new synergy between uncertainty-aware label reliability modeling and domain-category transfer modeling, which has not been explored in OSDG-NL. Unlike prior works that either (i) separate clean/noisy samples using feature-space prototypes (HyProMeta) or (ii) rely on linear interpolation (MixUp) for augmentation, our method introduces a fundamentally different paradigm. First, we propose Unsupervised Two-Stage Evidential Loss Clustering (UTS-ELC), which leverages evidential loss trajectories to capture not only prediction errors but also their associated uncertainties, enabling more reliable clean/noisy separation across domains. Second, we introduce Domain and Category Conditioned Residual Flow Matching (DC-CRFM), a flow-matching strategy conditioned on domain and category labels, which learns structured residuals rather than interpolations, thereby modeling diverse transfer paths between categories and domains. Finally, by integrating these two components within a meta-learning framework, we achieve principled decoupling of clean and noisy supervision, which is absent in prior methods. This combination enables EReLiFM to provide both uncertainty-aware noise diagnosis and diverse domain-category transfer modeling capabilities that neither clustering nor augmentation methods alone can offer. Our approach achieves state-of-the-art results on the PACS Li et al. (2017), DigitsDG Zhou et al. (2020a), and TerraINC Beery et al. (2018) datasets, showing its effectiveness in providing diverse cues to ensure correct optimization.

## 2 RELATED WORK

**Noisy Label Learning.** Accurate labels are crucial for deep learning models to acquire reliable information Xu et al. (2024), while mislabeled data can mislead the optimization Cheng et al. (2020). To combat label noise, various strategies have been proposed: label corruption probabilities modeling Xia et al. (2019); Tanno et al. (2019); Zhu et al. (2021b; 2022); Li et al. (2022), re-weighting samples to adjust loss contributions Liu & Tao (2016), and detecting noisy labels before training Song et al. (2019); Wei et al. (2022); Chen et al. (2021). TCL Huang et al. (2023) applies contrastive learning and Gaussian Mixture Models. Furthermore, noise-robust loss functions Liu & Guo (2020); Ma et al. (2020); Zhu et al. (2021a) and regularization tricks Wei et al. (2021); Cheng et al. (2023); Liu et al. (2022) enhance model resilience. Methods like BadLabel Zhang et al. (2024) and LSL Kim et al. (2024) leverage label-flipping attacks and label structure, respectively. Notably, HyProMeta Peng et al. (2024a) first introduces two benchmarks for the challenging OSDG-NL.

**Open-Set Domain Generalization.** Open-Set Domain Generalization (OSDG) presents two interrelated challenges: domain generalization Wang et al. (2020); Nam et al. (2021); Zhou et al. (2020c); Guo et al. (2023); Zhou et al. (2020b); Li et al. (2021a;b); Dong et al. (2024b), which trains models to transfer across source domains and the unseen, and open-set recognition Wang et al. (2024); Zhao et al. (2023); Bao et al. (2021); Geng et al. (2021); Peng et al. (2024c), which aims to reject unknown categories with low confidence scores Fu et al. (2020); Singha et al. (2024); Bose et al. (2023); Chen et al. (2022); Li et al. (2018); Zhao & Shen (2022). Although typically studied separately, OSDG explores strategies to address both challenges simultaneously. Previous work has investigated metric learning Katsumata et al. (2021), domain-augmented meta-learning Shu et al. (2021), and GAN-based data synthesis Bose et al. (2023) to boost model robustness. Recently, formalized OSDG protocols Wang et al. (2023) have demonstrated the effectiveness of meta-learning in handling OSDG. HyProMeta Peng et al. (2024a) focuses on hyperbolic prototypes to distinguish label-clean/noisy data, but is limited by the information scarcity of the limited label-clean samples. Multi-modal open set domain generalization task is for the first time proposed by Dong et al. (2024a). Gupta et al. (2025) explore Low-Shot Open-Set Domain Generalization (LSOSDG) task and propose masked cross-modal translation and multi-modal Jigsaw puzzle to achieve self-supervision. Flow-matching-based approaches Dao et al. (2023); Gat et al. (2024); Klein et al. (2023); Chen & Lipman (2023); Eijkelboom et al. (2024) have gained attention for their effectiveness in optimal transportation between distributions and real-world applications. We propose EReLiFM, which integrates

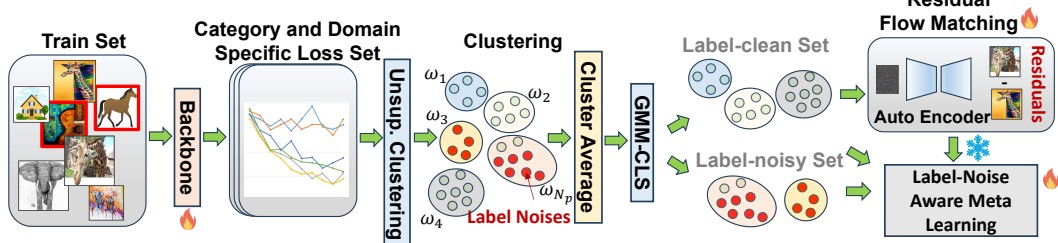

Figure 1: An overview of our proposed method. We first train the backbone and record the epoch-wise loss per sample. We cluster these losses into $N_p$ groups without a predefined number. Cluster averaging yields a new loss set per domain and category. GMM then performs binary separation, identifying the lower-loss cluster as the label-clean set, which trains the residual-conditioned flow matching to generate domain and categorical residuals. Finally, the partitioned dataset and trained model are integrated into our label noise aware meta learning (detailed in Alg. 1).

evidential-loss-based clean/noisy partitioning with domain- and category-conditioned residual flow in a meta-learning framework, achieving significant improvements over existing OSDG-NL methods.

## 3 METHODOLOGY

### 3.1 TASK DESCRIPTION

In this task, we consider a set containing $N_d$ domains $\mathcal{D} = \{d_1, d_2, ..., d_{N_d}\}$ and adopt the leave-one-out setting from Wang et al. (2023), where a single domain $d_t$ is reserved for testing, while the remaining $\mathcal{D}_S = \mathcal{D}/\{d_t\}$ serve as source domains during training. The dataset's label set $\mathcal{Y}$ consists of $\mathcal{Y}_k$ (known categories in training) and $\mathcal{Y}_u$ (unseen categories in test), where $\mathcal{Y} = \mathcal{Y}_k \cup \mathcal{Y}_u$. For each pair of the sample $\mathbf{x}_s$ and label $\mathbf{y}_s$ in the source domain, $\mathbf{y}_s$ is converted to other known categories according to the different label noise settings to simulate the annotation error. Our aim is to achieve the best optimization when label noise exists in open-set domain generalization.

### 3.2 EReLiFM

In this work, we propose Evidential Reliability-Aware Residual Flow Meta-Learning (EReLiFM) to deal with noisy labels within the realm of OSDG, which will be elaborated in this subsection.

Our method addresses OSDG under noisy labels through a three-stage design that improves data reliability, diversity, and supervision quality. First, we separate clean from noisy samples using UTS-ELC, which relies on evidential-loss trajectories and uncertainty rather than embedding similarity, enabling a more reliable partition under domain shift. Next, we enrich and diversify the clean subset with DC-CRFM, a flow-based residual modeling approach that synthesizes realistic cross-domain and cross-category variations to expand the effective training distribution. Finally, we optimize a meta-learning objective that decouples clean and noisy supervision: clean and augmented samples drive the meta-train updates, while noisy samples are handled in meta-test using evidential pseudo-labeling to prevent overfitting to incorrect annotations. Together, these stages form a coherent filter–enrich–decouple pipeline that achieves robust generalization in the presence of substantial label noise. The whole workflow of our proposed approach is depicted in Figure 1.

**Unsupervised Two-Stage Evidential Loss Clustering.** OSDG leverages reliable cues from source domains and known categories to recognize unknown categories in unseen domains Peng et al. (2024b); Wang et al. (2023). However, since label noise reduces the scale of reliable data, most of the existing works in the open-set domain generalization field deliver limited performance under label noise. Recognition of the data with label noise is critical to handle label noise for providing reliable optimization direction guidance for the deep learning model during optimization. Existing work, *i.e.*, HyProMeta Peng et al. (2024a), relies on clustering on embeddings for label noise agnostic learning, which is sensitive to outliers and feature quality, delivering limited performance. In this work, we optimize this process by proposing Unsupervised Two-Stage Evidential Loss Clustering (**UTS-ELC**), which separates label-clean/noisy data from a training dynamics perspective. We rely on evidential training dynamics instead of embeddings to achieve label noise diagnosis to avoid the

sensitivity to outlier embeddings. Early works Han et al. (2018); Liu et al. (2021) adopt multi-model joint optimization strategies based on the small-loss criterion. However, such approaches do not explicitly achieve a clear separation between clean and noisy labels. More recently, Yue & Jha (2024) introduce an unsupervised clustering strategy on recorded training dynamics to perform this separation. In contrast, our experiments show that a domain- and category-aware evidential loss leads to a more reliable distinction between clean and noisy sets under the open-set domain generalization scenario. Evidential loss models both evidence and uncertainty, providing a clearer training signal for separating clean and noisy samples. Clean samples quickly accumulate high evidence and low uncertainty, producing stable, low-loss trajectories. Noisy samples yield inconsistent evidence, leading to higher uncertainty and more volatile losses. Because evidential learning penalizes both errors and unwarranted confidence, mislabeled samples incur larger, more persistent penalties. In contrast, standard cross-entropy lacks an uncertainty term and cannot reliably distinguish samples whose losses overlap or fluctuate early in training. Optimized as a Dirichlet-based belief update, evidential loss pushes clean samples toward high-evidence regions while noisy samples stay in low-evidence regimes, creating a geometric margin in trajectory space that clustering methods can exploit. Next, we describe how to achieve UTS-ELC in detail.

To mitigate the detrimental impact of label noise on the residual flow matching design, we first categorize the data based on their recorded evidential loss trajectories, as samples trained with incorrect labels typically exhibit higher loss, in accordance with Co-Teaching Han et al. (2018). While UTS-ELC builds on the intuition of loss trajectory clustering, our key novelty is the use of evidential uncertainty and domain/category-specific cues, which provide a more reasonable and reliable separation of clean/noisy data under OSDG.

Initially, we train the backbone network on the entire dataset, despite the presence of label noise, while employing cyclic learning rates to improve convergence stability. Furthermore, we integrate evidential learning to enhance the model's generalization capability as Eq. (1). Evidential learning enables models to estimate both predictions and their associated uncertainty, leading to more reliable and calibrated predictions.

$$\mathcal{L}_{EL} = \sum_{i=1}^{\mathcal{C}} \left[ \mathbf{y}_i \left( \log S_{EL} - \log(\mathbf{M}_\alpha(\mathbf{x})_i + 1) \right) \right], \tag{1}$$

where $S_{EL} = \sum_{i=1}^{\mathcal{C}}(\mathrm{Dir}(p_{pred}|\mathbf{M}_\alpha(\mathbf{x})_i + 1))$ denotes the strength of a Dirichlet distribution, $\mathbf{M}_\alpha$ indicates the backbone, $\mathbf{y}_i$ is the one-hot annotation of sample $\mathbf{x}$ from class $i$, $p_{pred}$ is the predicted probability, and $\mathcal{C}$ is the class number.

During training, the evidential learning loss is recorded for each sample at every epoch. For a given sample $\mathbf{x}$, the recorded loss is represented as $\mathbf{l} = [l_1, l_2, ..., l_{N_e}]$, where $N_e$ denotes the total number of epochs. We then construct a new feature set based on the recorded losses for each sample, formulated as $\mathcal{X} = \{\mathbf{l}_i | \mathbf{x}_i \in \mathcal{T}\}$, where $\mathcal{T}$ represents the entire training set. To differentiate samples with and without label noise, we apply the unsupervised clustering method, FINCH Sarfraz et al. (2019), to the loss feature set, facilitating an initial hierarchical clustering process, according to Eq. (2) and Eq. (3).

$$\Omega_d^y = \{\omega_1, \omega_2, ..., \omega_{N_p}\} \leftarrow \mathrm{FINCH}(\mathcal{X}_d^y), \tag{2}$$

$$\Omega = \{\Omega_d^y | d \in \mathcal{D}_S, y \in \mathcal{Y}_k\}, \tag{3}$$

where $\Omega$ represents the complete set of partitions, and $\mathcal{X}_d^y$ and $\Omega_d^y$ denote the loss set and the set of unsupervised cluster partitions for domain $d$ and class $y$, respectively.

Next, we construct a new set by computing the average of the samples within each domain and category for each partition according to Eq. 4.

$$\hat{\mathcal{X}}_d^y = \{\mu(\omega_1(\mathcal{X}_d^y)), \mu(\omega_2(\mathcal{X}_d^y)), ..., \mu(\omega_{N_p}(\mathcal{X}_d^y))\}, \tag{4}$$

where $\mu(\cdot)$ denotes the averaging operation, $N_p$ denotes the total number of partitions clustered by the first level results on FINCH Sarfraz et al. (2019), and $\hat{\mathcal{X}}_d^y$ represents the resultant score set.

Finally, a Gaussian Mixture Model (GMM) based classifier (with two Gaussian components) is applied to perform a binary classification on the score set. This facilitates a threshold-free partitioning of the training data. The GMM class with the lower average loss is identified as the label-clean set (*i.e.*, $\hat{\mathcal{X}}_d^{(y,c)}$) and the other is denoted as noisy set (*i.e.*, $\hat{\mathcal{X}}_d^{(y,n)}$), as Eq. (5).

$$\hat{\mathcal{X}}_d^{(y,c)}, \hat{\mathcal{X}}_d^{(y,n)} = \mathrm{GMM}(\hat{\mathcal{X}}_d), s.t. \ \mu(\hat{\mathcal{X}}_d^{(y,c)}) < \mu(\hat{\mathcal{X}}_d^{(y,n)}). \tag{5}$$

---

**Algorithm 1** Training with EReLiFM.

---

**Require:** $\mathcal{D}_S$: source domain set; $\mathcal{Y}_k$: known category set; $\mathbf{M}_\alpha$: neural network backbone; $\mathbf{M}_\gamma$: flow matching model; $\mathcal{L}_{CE}$: cross entropy loss; $\mathcal{L}_{EL}$: evidential learning loss; $\mathcal{T}$: dataset with label noise; $\mathbf{r}_0$: random Gaussian noise.

1: Dataset separation, $\mathcal{T}_{clean}, \mathcal{T}_{noisy} \leftarrow$ **UTS-ELC**$(\mathcal{T})$.
2: Train $\mathbf{M}_\gamma$ on $\mathcal{T}_{clean}$ using domain and category residuals, conditioned by classes and domains.
3: **while** not converged **do**
4:  ▷ *Meta-Training Stage*                   ◁
5:  $\mathbf{B}_{clean} \leftarrow Iter(\mathcal{T}_{clean})$, with domain label and category label $\mathbf{y}_d$ and $\mathbf{y}_c$.
6:  Sample $\hat{\mathbf{y}}_c \leftarrow \mathcal{Y}/\{\mathbf{y}_c\}$, and $\hat{\mathbf{y}}_d \leftarrow \mathcal{D}_S/\{\mathbf{y}_d\}$.
7:  $\mathbf{R}_d \leftarrow \mathbf{M}_\gamma(\mathbf{r}_0, (\mathbf{y}_c, \mathbf{y}_c), (\mathbf{y}_d, \hat{\mathbf{y}}_d))$, generate domain residual.
8:  $\mathbf{B}_{dr} \leftarrow Add(\mathbf{B}_{clean}, \mathbf{R}_d)$, merge domain residual.
9:  Assign $\mathbf{y}_c \rightarrow \mathbf{y}_{dr}$ for $\mathbf{B}_{dr}$.
10:  $\mathbf{R}_c \leftarrow \mathbf{M}_\gamma(\mathbf{n}_0, (\mathbf{y}_c, \hat{\mathbf{y}}_c), (\mathbf{y}_d, \mathbf{y}_d))$, generate category residual.
11:  $\mathbf{B}_{cr} \leftarrow Add(\mathbf{B}_{clean}, \mathbf{R}_c)$, merge categorical residual.
12:  Assign $\mathbf{y}_a \rightarrow \mathbf{y}_{cr}$ for $\mathbf{B}_{cr}$, where $\mathbf{y}_a$ denotes an additional class beyond known classes.
13:  Update parameters based on $\mathcal{L}_{m-train} = \lambda_c * \mathcal{L}_{CE}(\mathbf{B}_{clean}, \mathbf{y}_c) + \lambda_{dr} * \mathcal{L}_{CE}(\mathbf{B}_{dr}, \mathbf{y}_{dr}) + \lambda_{cr} * \mathcal{L}_{CE}(\mathbf{B}_{cr}, \mathbf{y}_{cr})$.
14:  ▷ *Meta-Test Stage*                     ◁
15:  $\mathbf{B}_{noisy} \leftarrow Iter(\mathcal{T}_{noisy})$ with category label $\mathbf{y}_{nc}$.
16:  $\mathbf{y}_{pseudo} = ArgMax(\mathbf{M}_\alpha(\mathbf{B}_{noisy}))$.
17:  $\mathcal{L}_{m-test} = \lambda_p * \mathcal{L}_{EL}(\mathbf{B}_{noisy}, \mathbf{y}_{pseudo}) + \lambda_{nc} * \mathcal{L}_{CE}(\mathbf{B}_{noisy}, \mathbf{y}_{nc})$.
18:  UpdateParameters$(\mathcal{L}_{m-test} + \mathcal{L}_{m-train})$.     *// Final Parameter Update*

---

We then obtain the corresponding dataset according to the aforementioned partition manner, where we use $\mathcal{T}_{clean}$ and $\mathcal{T}_{noisy}$ to denote the clean set and noisy set, respectively.

**Domain and Category Conditioned Residual Flow Matching.** Despite the aforementioned evidential loss-based separation strategy, training remains challenged by the scarcity of reliably annotated data. HyProMeta Peng et al. (2024a) addresses this issue through cross-category MixUp and learnable prompts, thereby expanding the data scope to stabilize training. Yet, the diversity remains limited, since MixUp models only a single interpolation path between source and target data. To mitigate domain shift, enhance the model's sensitivity to diverse category transfers, and expand the scale of reliably annotated training data, we introduce **D**omain and **C**ategory **C**onditioned **R**esidual **F**low **M**atching (DC-CRFM). DC-CRFM generates diverse transfer paths by reconstructing domain- and category-residuals from random noise, conditioned on both domain and category labels. In this way, DC-CRFM explicitly models transitions across categories and domains, boosting generalization during training. Importantly, we train DC-CRFM on the clean subset identified by **UTS-ELC**. Unlike MixUp, which interpolates between samples, DC-CRFM learns structured residuals across domains and categories. As demonstrated in our ablations (Tab. 7), this design yields significant improvements over MixUp, evidencing that DC-CRFM is fundamentally distinct from interpolation-based augmentation.

Flow matching is a technique in machine learning that aligns feature distributions between source and target domains Lipman et al. (2023). It appears as an efficient alternative compared with diffusion models Ho et al. (2020) for data generation, where methods leveraging straight flows are introduced by Liu et al. (2023).

In our work, we propose a domain and category conditioned residual flow matching strategy to enrich the paths across different domains and categories based on a clean label set $\mathcal{T}_{clean}$. Domain residuals represent the visual differences between samples of the same category from different domains, while category residuals capture discrepancies between different categories within the same domain. We use our proposed conditioned flow matching to generate category and domain residuals.

To enrich cross-domain and cross-category transfer, we propose **D**omain and **C**ategory **C**onditioned **R**esidual **F**low **M**atching (DC-CRFM), a conditioned variant of flow matching that learns residual distributions rather than directly generating samples. Given a source sample $\mathbf{I}_s$, a target sample $\mathbf{I}_t$ and a condition $\mathbf{q}$ (*e.g.*, source→target domain/category pair), RFM draws a residual $\mathbf{r}_1 = \mathbf{I}_t - \mathbf{I}_s \sim p_r^{(\mathbf{q})}$

via a probability-flow ODE driven by a conditioned vector field $f_\theta$. Training is depicted in Eq. 6.

$$\mathcal{L}_{\text{RFM}} = \mathbb{E}_{(\mathbf{q},\, \mathbf{r}_0,\, \mathbf{r}_1,\, t)} \big[ \| f_\theta(\mathbf{r}_t, t, \psi(\mathbf{q})) - (\mathbf{r}_1 - \mathbf{r}_0) \|_2^2 \big], \quad \mathbf{r}_t = (1-t)\mathbf{r}_0 + t\mathbf{r}_1, \tag{6}$$

where $\mathbf{r}_0 \sim \mathcal{N}(0,1)$ (Gaussian distribution), $\mathbf{r}_1 \sim p_r^{(\mathbf{q})}$, $t \sim \mathcal{U}(0,1)$ (normal distribution), and $\psi(\mathbf{q})$ encodes the condition. At inference, integrating $\frac{d\mathbf{r}}{dt} = f_\theta(\mathbf{r}, t, \psi(\mathbf{q}))$ from noise $\mathbf{r}_0 \sim \mathcal{N}(0,1)$ yields $\mathbf{r} \sim p_r^{(\mathbf{q})}$, which is then added to $\mathbf{I}_s$ to form an augmented sample $\mathbf{I}_{\text{aug}} = \mathbf{I}_s + \mathbf{r}$. This design captures structured residual transitions between domains and categories, in contrast to simple interpolations such as MixUp Zhou et al. (2020c); Peng et al. (2024a).

**Evidential Reliability-Aware Residual Flow Meta-Learning.** Meta-learning has been proven effective for open-set domain generalization by constructing tailored meta-tasks to promote cross-domain generalization Wang et al. (2023); Peng et al. (2024b;a). Building upon this insight, our main training framework adopts a meta-learning paradigm. Specifically, we define a new meta-training task over UTS-ELC-selected clean data and DC-CRFM-augmented clean data based on UTS-ELC selection. The optimized model from meta-training is then used to improve optimization on the noisy subset during meta-testing. Here, DC-CRFM plays a central role by enriching label-clean data with diverse category/domain transfer paths. Samples from the noisy set are supervised with high-confidence pseudo-labels via evidential learning, and regularized by cross-entropy loss against the original labels, thereby reinforcing consistency with the label-clean set. Compared with HyProMeta Peng et al. (2024a), our meta-task differs in both meta-train and meta-test phases. In the meta-train stage, we exclusively rely on DC-CRFM augmented clean data, avoiding any optimization over noisy samples. In the meta-test stage, we focus solely on the noisy set: pseudo-labels are assigned via maximum-confidence predictions, and supervision is defined by a competition between pseudo-labels and original labels, as UTS-ELC does not guarantee perfect separation. To further account for uncertainty, evidential supervision is imposed on the pseudo-labels.

Through this process, we obtain a flow matching model that generates domain and category residuals while distinguishing clean from noisy data. These components are integrated into meta-learning for denoising and improving generalization in OSDG, as outlined in Alg. 1.

We first separate clean and noisy data using UTS-ELC, then train $\mathbf{M}_\alpha$ on $\mathcal{T}_{clean}$ with residual augmentation. In the meta-train stage, for each $\mathbf{B}_{clean}$ with labels $(\mathbf{y}_c, \mathbf{y}_d)$, we sample $(\hat{\mathbf{y}}_c, \hat{\mathbf{y}}_d)$ to generate residuals, producing $\mathbf{B}_{dr}$ (domain residual, supervised by $(\mathbf{y}_c, \mathbf{y}_d)$) and $\mathbf{B}_{cr}$ (category residual, supervised by an additional class $\mathbf{y}_a$). We assign $\mathbf{y}_a \to \mathbf{y}_{cr}$ for $\mathbf{B}_{cr}$. The model is updated with $\mathcal{L}_{m-train} = \lambda_c * \mathcal{L}_{CE}(\mathbf{B}_{clean}, \mathbf{y}_c) + \lambda_{dr} * \mathcal{L}_{CE}(\mathbf{B}_{dr}, \mathbf{y}_{dr}) + \lambda_{cr} * \mathcal{L}_{CE}(\mathbf{B}_{cr}, \mathbf{y}_{cr})$.

In the meta-test stage, noisy samples $\mathbf{B}_{noisy}$ are optimized via competition between the original label $\mathbf{y}_{nc}$ and a pseudo-label $\mathbf{y}_{pseudo} = ArgMax(M_\alpha(\mathbf{B}_{noisy}))$, with evidential regularization. The meta-test loss is calculated as $\mathcal{L}_{m-test} = \lambda_p * \mathcal{L}_{EL}(\mathbf{B}_{noisy}, \mathbf{y}_{pseudo}) + \lambda_{nc} * \mathcal{L}_{CE}(\mathbf{B}_{noisy}, \mathbf{y}_{nc})$.

An auxiliary cross-entropy term ensures that useful cues can still be extracted from misclassified clean samples. The final loss combines both stages, $\mathcal{L}_{m-train} + \mathcal{L}_{m-test}$, ensuring robust optimization with reliable supervision. This pipeline strengthens cross-domain generalization while also improving recognition of out-of-distribution categories. Overall, clean/noisy separation via evidential training dynamics enables reliable residual flow training, while flow-augmented clean data and noisy samples are optimized separately in meta-train and meta-test to ensure robust learning.

## 4 EXPERIMENTS

### 4.1 NOISY LABEL SETTINGS

We adopt the setting of HyProMeta Peng et al. (2024a) for OSDG-NL, incorporating **symmetric** and **asymmetric** label noise. **Symmetric noise** randomly reassigns class labels at predefined rates ($20\%$, $50\%$, $80\%$) without considering semantics. In contrast, **asymmetric noise** mislabels samples according to semantic similarity using BERT Devlin et al. (2019) for textual feature extraction and cosine similarity for class similarity computation. The asymmetric noise level is set to $50\%$.

| Method | Photo (P) | | | Art (A) | | | Cartoon (C) | | | Sketch (S) | | | Avg | | |
|---|---|---|---|---|---|---|---|---|---|---|---|---|---|---|---|
| | Acc | H-score | OSCR | Acc | H-score | OSCR | Acc | H-score | OSCR | Acc | H-score | OSCR | Acc | H-score | OSCR |
| TCL Huang et al. (2023) | 58.32 | 59.21 | 51.72 | 53.66 | 48.28 | 42.91 | 46.78 | 38.29 | 31.74 | 31.55 | 22.88 | 24.30 | 47.58 | 42.17 | 37.67 |
| NPN Sheng et al. (2024) | 64.30 | 70.87 | 61.99 | 51.66 | 52.10 | 45.40 | 55.65 | 44.88 | 38.64 | 35.58 | 22.35 | 25.86 | 51.80 | 47.55 | 42.97 |
| BadLabel Zhang et al. (2024) | 54.93 | 55.73 | 48.24 | 53.72 | 53.25 | 46.55 | 50.23 | 55.36 | 45.70 | 31.55 | 21.84 | 28.38 | 47.61 | 46.55 | 42.22 |
| DISC Li et al. (2023) | 53.47 | 56.13 | 47.22 | 54.47 | 47.46 | 43.48 | 53.27 | 53.97 | 44.33 | 24.01 | 16.75 | 11.52 | 46.31 | 43.58 | 36.64 |
| LSL Kim et al. (2024) | 58.97 | 58.93 | 52.15 | 49.97 | 48.17 | 39.20 | 47.50 | 44.07 | 34.63 | 30.59 | 12.44 | 16.81 | 46.76 | 40.90 | 35.70 |
| PLM Zhao et al. (2024) | 55.57 | 42.33 | 38.27 | 41.78 | 43.09 | 32.95 | 45.75 | 40.44 | 33.26 | 33.27 | 12.11 | 15.46 | 38.33 | 34.49 | 29.99 |
| ARPL Bendale & Boult (2016) | 62.52 | 67.96 | 59.46 | 52.35 | 45.29 | 41.09 | 50.13 | 44.47 | 37.01 | 29.56 | 13.49 | 22.70 | 48.64 | 42.80 | 40.07 |
| ODGNet Bose et al. (2023) | 63.00 | 70.61 | 61.18 | 58.08 | 40.01 | 44.97 | 58.33 | 53.37 | 48.89 | 22.84 | 9.69 | 16.48 | 50.56 | 43.42 | 42.88 |
| MLDG Shu et al. (2019) | 60.26 | 69.11 | 59.35 | 58.66 | 55.83 | 49.03 | 58.07 | 51.18 | 45.08 | 25.87 | 18.48 | 16.40 | 37.22 | 48.65 | 42.47 |
| SWAD Cha et al. (2021) | 59.94 | 69.23 | 58.69 | 49.59 | 48.04 | 40.04 | 37.44 | 34.32 | 25.96 | 19.10 | 20.72 | 12.86 | 41.52 | 41.52 | 34.39 |
| MixStyle Zhou et al. (2020c) | 60.10 | 65.39 | 56.89 | 55.16 | 44.70 | 44.01 | 59.31 | 47.35 | 39.93 | 34.54 | 17.49 | 20.86 | 52.28 | 43.73 | 40.42 |
| MEDIC-cls Wang et al. (2023) | 62.20 | 52.63 | 53.23 | 54.60 | 54.05 | 46.51 | 59.31 | 52.02 | 47.65 | 34.54 | 28.22 | 21.44 | 52.66 | 46.73 | 41.96 |
| MEDIC-bcls Wang et al. (2023) | 62.20 | 57.47 | 53.93 | 54.60 | 53.10 | 46.38 | 59.31 | 53.70 | 48.68 | 34.54 | 32.71 | 24.06 | 52.66 | 49.25 | 42.76 |
| EBiL-HaDS-cls Peng et al. (2024b) | 65.19 | 58.09 | 57.84 | 53.28 | 47.07 | 40.36 | 57.56 | 52.17 | 45.95 | 37.52 | 28.83 | 22.31 | 53.39 | 46.54 | 41.62 |
| EBiL-HaDS-bcls Peng et al. (2024b) | 65.19 | 63.82 | 60.63 | 53.28 | 46.70 | 39.80 | 57.56 | 50.58 | 45.63 | 37.52 | 30.61 | 26.55 | 53.39 | 47.93 | 43.15 |
| HyProMeta Peng et al. (2024a) | 66.00 | 76.84 | 66.00 | 59.91 | 56.89 | 49.93 | 59.41 | 56.47 | 50.42 | 39.16 | 34.76 | 26.46 | 56.12 | 56.24 | 48.20 |
| Ours | 82.39 | 81.52 | 78.68 | 77.61 | 66.14 | 65.37 | 65.39 | 55.26 | 65.39 | 58.11 | 44.56 | 38.15 | 70.88 | 61.87 | 61.90 |

Table 1: Results (%) of PACS on ResNet18. The open-set ratio is 6:1 and symmetric label noise is with ratio 20%.

| Method | Photo (P) | | | Art (A) | | | Cartoon (C) | | | Sketch (S) | | | Avg | | |
|---|---|---|---|---|---|---|---|---|---|---|---|---|---|---|---|
| | Acc | H-score | OSCR | Acc | H-score | OSCR | Acc | H-score | OSCR | Acc | H-score | OSCR | Acc | H-score | OSCR |
| TCL Huang et al. (2023) | 54.68 | 52.40 | 46.51 | 52.78 | 22.65 | 30.94 | 47.19 | 37.73 | 34.89 | 26.33 | 9.83 | 7.62 | 45.25 | 30.65 | 29.99 |
| NPN Sheng et al. (2024) | 48.38 | 38.12 | 33.55 | 35.71 | 32.33 | 24.47 | 38.94 | 26.88 | 18.60 | 26.93 | 27.59 | 18.96 | 37.49 | 31.23 | 23.90 |
| BadLabel Zhang et al. (2024) | 46.20 | 57.45 | 45.07 | 45.34 | 47.29 | 37.89 | 35.17 | 43.35 | 32.14 | 28.40 | 26.95 | 15.71 | 38.78 | 43.76 | 32.70 |
| DISC Li et al. (2023) | 52.52 | 56.07 | 50.55 | 46.84 | 31.91 | 30.35 | 28.47 | 28.28 | 19.97 | 30.83 | 25.63 | 24.78 | 39.67 | 35.47 | 31.41 |
| LSL Kim et al. (2024) | 41.36 | 30.83 | 20.27 | 42.28 | 39.78 | 31.40 | 42.39 | 37.59 | 30.89 | 26.90 | 15.40 | 7.42 | 38.23 | 30.90 | 22.50 |
| PLM Zhao et al. (2024) | 55.57 | 42.33 | 38.27 | 39.21 | 27.81 | 24.93 | 33.01 | 27.81 | 21.49 | 25.52 | 6.65 | 13.41 | 38.33 | 26.15 | 24.53 |
| ARPL Bendale & Boult (2016) | 55.41 | 62.40 | 54.17 | 45.72 | 44.50 | 34.51 | 43.73 | 38.44 | 30.13 | 27.30 | 7.65 | 20.95 | 43.04 | 38.25 | 34.94 |
| ODGNet Bose et al. (2023) | 60.66 | 63.57 | 56.75 | 55.09 | 40.01 | 44.97 | 46.52 | 39.85 | 32.10 | 32.02 | 24.40 | 17.09 | 48.57 | 41.96 | 37.73 |
| MLDG Shu et al. (2019) | 59.37 | 68.02 | 58.54 | 56.49 | 50.15 | 44.92 | 46.78 | 46.02 | 36.91 | 23.69 | 24.32 | 16.40 | 46.58 | 47.13 | 39.19 |
| SWAD Cha et al. (2021) | 58.58 | 67.77 | 56.25 | 45.78 | 41.39 | 38.30 | 34.19 | 33.89 | 23.95 | 20.43 | 14.15 | 6.81 | 39.75 | 39.30 | 31.33 |
| MixStyle Zhou et al. (2020c) | 54.04 | 62.25 | 30.23 | 41.78 | 37.68 | 27.03 | 47.09 | 26.67 | 27.03 | 30.88 | 22.81 | 17.09 | 43.45 | 37.35 | 25.35 |
| MEDIC-cls Wang et al. (2023) | 60.58 | 51.37 | 44.29 | 53.28 | 51.88 | 44.12 | 50.54 | 49.07 | 42.84 | 36.67 | 28.00 | 20.83 | 50.27 | 45.08 | 38.02 |
| MEDIC-bcls Wang et al. (2023) | 60.58 | 48.99 | 43.25 | 53.28 | 37.32 | 33.99 | 50.54 | 44.08 | 39.39 | 36.67 | 30.58 | 21.83 | 50.27 | 40.24 | 34.62 |
| EBiL-HaDS-cls Peng et al. (2024b) | 61.15 | 62.20 | 54.97 | 52.47 | 43.90 | 37.71 | 49.66 | 48.05 | 40.75 | 34.39 | 28.62 | 20.98 | 49.42 | 45.69 | 38.60 |
| EBiL-HaDS-bcls Peng et al. (2024b) | 61.15 | 25.32 | 48.79 | 52.47 | 42.61 | 36.20 | 49.66 | 49.13 | 41.34 | 34.39 | 21.33 | 21.70 | 49.42 | 34.60 | 37.00 |
| HyProMeta Peng et al. (2024a) | 61.15 | 73.38 | 63.79 | 60.85 | 52.13 | 46.97 | 51.99 | 49.44 | 41.71 | 39.06 | 33.53 | 23.44 | 54.27 | 52.24 | 43.98 |
| Ours | 81.91 | 78.14 | 77.52 | 70.29 | 61.86 | 59.58 | 61.89 | 50.00 | 45.53 | 49.22 | 37.69 | 29.34 | 65.83 | 56.92 | 52.99 |

Table 2: Results (%) of PACS on ResNet18. The open-set ratio is 6:1 and symmetric label noise is with ratio 50%.

| Method | Photo (P) | | | Art (A) | | | Cartoon (C) | | | Sketch (S) | | | Avg | | |
|---|---|---|---|---|---|---|---|---|---|---|---|---|---|---|---|
| | Acc | H-score | OSCR | Acc | H-score | OSCR | Acc | H-score | OSCR | Acc | H-score | OSCR | Acc | H-score | OSCR |
| TCL Huang et al. (2023) | 31.58 | 25.81 | 17.39 | 27.08 | 26.60 | 16.45 | 27.69 | 27.17 | 17.66 | 21.20 | 8.52 | 14.87 | 26.89 | 22.03 | 16.59 |
| NPN Sheng et al. (2024) | 17.21 | 12.49 | 10.18 | 24.27 | 12.67 | 13.87 | 22.85 | 12.99 | 10.85 | 19.66 | 4.31 | 11.63 | 21.00 | 10.62 | 11.63 |
| BadLabel Zhang et al. (2024) | 22.62 | 14.11 | 22.62 | 15.95 | 14.30 | 10.16 | 19.39 | 24.34 | 14.95 | 26.13 | 14.14 | 17.96 | 21.02 | 16.77 | 16.42 |
| DISC Li et al. (2023) | 22.05 | 19.53 | 15.27 | 24.77 | 23.65 | 17.53 | 27.13 | 22.49 | 14.76 | 16.03 | 12.86 | 10.00 | 22.05 | 19.63 | 14.39 |
| LSL Kim et al. (2024) | 18.58 | 22.82 | 13.89 | 23.64 | 16.71 | 14.56 | 15.37 | 15.84 | 7.94 | 21.68 | 1.92 | 8.26 | 19.82 | 14.32 | 11.16 |
| PLM Zhao et al. (2024) | 24.39 | 8.71 | 9.41 | 30.08 | 24.67 | 17.85 | 20.94 | 13.16 | 12.21 | 21.76 | 21.26 | 11.74 | 24.29 | 16.95 | 12.80 |
| ARPL Bendale & Boult (2016) | 38.77 | 23.79 | 15.88 | 22.12 | 20.58 | 11.40 | 23.98 | 14.45 | 8.98 | 25.76 | 16.45 | 11.71 | 27.66 | 18.82 | 11.99 |
| ODGNet Bose et al. (2023) | 31.18 | 19.56 | 18.49 | 27.64 | 6.64 | 12.81 | 20.78 | 21.43 | 12.81 | 21.65 | 22.00 | 7.90 | 25.31 | 17.41 | 13.00 |
| MLDG Shu et al. (2019) | 33.04 | 9.18 | 12.11 | 22.45 | 19.28 | 11.18 | 28.16 | 23.38 | 13.64 | 23.19 | 4.47 | 16.40 | 26.71 | 14.08 | 13.33 |
| SWAD Cha et al. (2021) | 18.09 | 18.69 | 10.19 | 22.51 | 20.22 | 11.97 | 23.67 | 21.96 | 11.89 | 19.75 | 12.20 | 15.29 | 21.01 | 18.27 | 12.33 |
| MixStyle Zhou et al. (2020c) | 25.28 | 22.05 | 16.88 | 24.70 | 17.68 | 12.90 | 21.61 | 20.39 | 11.14 | 24.44 | 12.39 | 15.13 | 24.01 | 18.13 | 14.01 |
| MEDIC-cls Wang et al. (2023) | 30.61 | 15.03 | 21.20 | 22.33 | 22.47 | 14.20 | 29.55 | 26.02 | 14.96 | 23.11 | 15.61 | 8.74 | 26.40 | 19.78 | 14.78 |
| MEDIC-bcls Wang et al. (2023) | 30.61 | 12.82 | 11.92 | 22.33 | 21.15 | 11.45 | 29.55 | 22.67 | 13.82 | 23.11 | 8.34 | 7.47 | 26.40 | 16.25 | 11.17 |
| EBiL-HaDS-cls Peng et al. (2024b) | 40.06 | 39.36 | 34.58 | 19.51 | 3.89 | 5.30 | 29.40 | 26.25 | 18.44 | 25.44 | 23.73 | 18.01 | 28.60 | 23.31 | 19.08 |
| EBiL-HaDS-bcls Peng et al. (2024b) | 40.06 | 15.06 | 23.84 | 19.51 | 12.30 | 11.93 | 29.40 | 29.83 | 18.73 | 25.44 | 26.37 | 17.71 | 28.60 | 20.89 | 18.05 |
| HyProMeta Peng et al. (2024a) | 47.01 | 34.98 | 43.29 | 28.77 | 25.28 | 20.07 | 31.40 | 28.38 | 18.59 | 26.72 | 25.04 | 18.40 | 33.48 | 28.42 | 25.09 |
| Ours | 54.04 | 48.50 | 47.16 | 32.83 | 34.52 | 24.04 | 43.79 | 37.94 | 29.13 | 23.83 | 29.26 | 19.75 | 38.62 | 37.56 | 30.02 |

Table 3: Results (%) of PACS on ResNet18. The open-set ratio is 6:1 and symmetric label noise is with ratio 80%.

| Method | Photo (P) | | | Art (A) | | | Cartoon (C) | | | Sketch (S) | | | Avg | | |
|---|---|---|---|---|---|---|---|---|---|---|---|---|---|---|---|
| | Acc | H-score | OSCR | Acc | H-score | OSCR | Acc | H-score | OSCR | Acc | H-score | OSCR | Acc | H-score | OSCR |
| TCL Huang et al. (2023) | 15.83 | 3.33 | 10.69 | 35.21 | 30.54 | 20.96 | 26.46 | 24.24 | 14.88 | 20.35 | 1.20 | 2.63 | 24.46 | 14.83 | 12.29 |
| NPN Sheng et al. (2024) | 44.43 | 37.82 | 28.94 | 38.65 | 39.48 | 23.11 | 31.92 | 20.12 | 13.49 | 23.24 | 11.25 | 4.29 | 34.56 | 27.17 | 17.46 |
| BadLabel Zhang et al. (2024) | 37.16 | 44.56 | 35.70 | 26.89 | 31.98 | 24.77 | 31.10 | 34.52 | 25.33 | 13.88 | 16.21 | 20.38 | 27.26 | 31.82 | 26.54 |
| DISC Li et al. (2023) | 46.20 | 42.31 | 38.82 | 44.28 | 42.69 | 34.41 | 42.70 | 35.22 | 26.28 | 32.77 | 2.77 | 22.37 | 41.49 | 30.75 | 30.47 |
| LSL Kim et al. (2024) | 27.30 | 19.63 | 14.32 | 24.27 | 11.56 | 13.18 | 25.63 | 14.77 | 12.04 | 22.21 | 16.31 | 25.07 | 24.85 | 15.57 | 16.15 |
| PLM Zhao et al. (2024) | 17.37 | 23.20 | 14.13 | 25.45 | 24.44 | 15.04 | 20.06 | 9.68 | 11.93 | 20.48 | 2.60 | 3.24 | 20.84 | 14.98 | 11.09 |
| ARPL Bendale & Boult (2016) | 38.69 | 28.90 | 31.88 | 37.71 | 29.82 | 19.55 | 33.99 | 24.57 | 16.20 | 20.56 | 26.92 | 17.79 | 32.74 | 27.55 | 21.35 |
| ODGNet Bose et al. (2023) | 45.15 | 49.11 | 39.31 | 37.59 | 34.82 | 24.43 | 42.96 | 42.10 | 24.43 | 26.00 | 15.02 | 15.65 | 37.93 | 35.26 | 25.96 |
| MLDG Shu et al. (2019) | 51.21 | 40.25 | 45.11 | 42.21 | 32.24 | 26.34 | 44.46 | 38.66 | 30.80 | 24.36 | 1.32 | 15.99 | 40.56 | 28.12 | 29.56 |
| SWAD Cha et al. (2021) | 40.47 | 14.21 | 35.14 | 32.15 | 18.26 | 9.48 | 20.06 | 12.52 | 8.53 | 20.48 | 11.83 | 5.47 | 28.29 | 14.20 | 14.66 |
| MixStyle Zhou et al. (2020c) | 49.76 | 41.10 | 40.39 | 35.96 | 36.32 | 26.54 | 41.72 | 32.04 | 23.55 | 25.58 | 4.81 | 23.55 | 38.25 | 28.57 | 28.51 |
| MEDIC-cls Wang et al. (2023) | 46.20 | 45.01 | 37.13 | 37.46 | 29.69 | 22.92 | 36.41 | 30.80 | 20.83 | 31.07 | 23.88 | 13.29 | 37.79 | 32.35 | 23.54 |
| MEDIC-bcls Wang et al. (2023) | 46.20 | 48.97 | 39.86 | 37.46 | 20.65 | 17.95 | 36.41 | 32.06 | 22.69 | 31.07 | 25.23 | 15.23 | 37.79 | 31.73 | 23.93 |
| EBiL-HaDS-cls Peng et al. (2024b) | 54.93 | 56.65 | 48.44 | 31.58 | 30.11 | 21.19 | 37.13 | 32.23 | 22.83 | 25.44 | 23.73 | 18.01 | 37.27 | 35.68 | 27.62 |
| EBiL-HaDS-bcls Peng et al. (2024b) | 54.93 | 30.36 | 40.79 | 31.58 | 30.97 | 18.73 | 37.13 | 32.56 | 23.21 | 25.44 | 26.37 | 17.71 | 37.27 | 30.07 | 25.11 |
| HyProMeta Peng et al. (2024a) | 51.62 | 61.33 | 49.38 | 45.28 | 41.74 | 35.72 | 49.25 | 49.63 | 39.63 | 38.50 | 37.41 | 26.48 | 46.16 | 47.53 | 37.80 |
| Ours | 69.22 | 66.92 | 64.07 | 56.35 | 47.20 | 42.76 | 59.46 | 48.91 | 42.14 | 47.55 | 41.62 | 35.03 | 58.15 | 51.16 | 46.00 |

Table 4: Results (%) of PACS on ResNet18. The open-set ratio is 6:1 and asymmetric label noise is with ratio 50%.

## 4.2 DATASETS AND METRICS

We adopt OSDG protocols from MEDIC Wang et al. (2023) and HyProMeta Peng et al. (2024a), where training domains share the same categories. Evaluation is on three benchmarks: **PACS** Li et al. (2017) ( *photo*, *art-painting*, *cartoon*, *sketch*), **DigitsDG** Zhou et al. (2020a) (*mnist*, *mnist-m*, *svhn*, *syn*), and **TerraINC** Beery et al. (2018) (reported in Tab. 16 in appendix). We follow the

| Method | 20% sym Acc | H-score | OSCR | 50% sym Acc | H-score | OSCR | 80 % sym Acc | H-score | OSCR | 50% asym Acc | H-score | OSCR |
|---|---|---|---|---|---|---|---|---|---|---|---|---|
| TCL Huang et al. (2023) | 52.47 | 55.28 | 46.85 | 50.19 | 43.92 | 42.02 | 23.31 | 18.44 | 12.26 | 38.91 | 39.94 | 30.98 |
| NPN Sheng et al. (2024) | 47.68 | 44.04 | 38.51 | 32.06 | 29.07 | 23.83 | 17.95 | 11.12 | 11.42 | 25.54 | 17.72 | 13.20 |
| BadLabel Zhang et al. (2024) | 49.06 | 50.57 | 46.04 | 39.83 | 45.65 | 36.62 | 20.92 | 22.11 | 19.63 | 32.27 | 41.03 | 31.86 |
| DISC Li et al. (2023) | 52.21 | 40.11 | 42.93 | 36.73 | 34.36 | 27.93 | 22.77 | 9.89 | 12.67 | 28.99 | 13.55 | 11.21 |
| LSL Kim et al. (2024) | 52.96 | 48.54 | 49.12 | 50.19 | 42.60 | 40.80 | 23.39 | 12.55 | 12.69 | 35.84 | 23.88 | 19.73 |
| PLM Zhao et al. (2024) | 52.94 | 47.03 | 46.62 | 42.17 | 37.45 | 35.33 | 26.40 | 15.88 | 9.72 | 24.78 | 23.62 | 17.48 |
| ARPL Bendale & Boult (2016) | 55.35 | 48.84 | 48.35 | 45.28 | 39.26 | 36.52 | 21.62 | 16.48 | 14.68 | 38.81 | 36.58 | 30.08 |
| ODGNet Bose et al. (2023) | 54.89 | 50.24 | 48.36 | 55.12 | 53.43 | 48.31 | 20.92 | 15.08 | 8.27 | 40.69 | 39.08 | 31.92 |
| MLDG Shu et al. (2019) | 55.30 | 50.06 | 48.92 | 55.89 | 52.98 | 49.64 | 25.77 | 21.04 | 15.74 | 44.50 | 46.46 | 40.70 |
| SWAD Cha et al. (2021) | 53.59 | 54.58 | 48.98 | 52.51 | 55.16 | 47.03 | 23.75 | 19.62 | 13.90 | 43.34 | 43.34 | 33.77 |
| MixStyle Zhou et al. (2020c) | 53.00 | 45.66 | 43.59 | 41.20 | 33.14 | 30.27 | 24.00 | 14.97 | 15.35 | 43.56 | 39.84 | 36.88 |
| MEDIC-cls Wang et al. (2023) | 56.76 | 52.64 | 47.99 | 53.61 | 48.99 | 47.08 | 28.97 | 23.03 | 16.36 | 42.42 | 43.23 | 35.49 |
| MEDIC-bcls Wang et al. (2023) | 56.76 | 48.54 | 48.45 | 53.61 | 40.86 | 38.67 | 28.98 | 18.43 | 13.10 | 42.42 | 40.01 | 33.31 |
| EBiL-HaDS-cls Peng et al. (2024b) | 56.24 | 49.48 | 46.67 | 52.68 | 45.77 | 44.39 | 30.87 | 14.26 | 18.71 | 40.49 | 40.39 | 32.21 |
| EBiL-HaDS-bcls Peng et al. (2024b) | 56.24 | 47.26 | 46.50 | 52.68 | 38.91 | 35.68 | 30.87 | 25.35 | 16.90 | 40.49 | 38.72 | 27.29 |
| HyProMeta Peng et al. (2024a) | 59.65 | 60.06 | 54.97 | 58.68 | 59.33 | 52.91 | 37.06 | 29.09 | 25.26 | 49.99 | 48.47 | 43.44 |
| Ours | 73.07 | 61.00 | 61.24 | 71.80 | 60.79 | 63.21 | 41.63 | 36.75 | 34.96 | 55.05 | 55.61 | 50.60 |

Table 5: Results (%) of PACS on ViT-Base. The open-set ratio is 6:1. The average domain performance is reported.

| Method | 20% sym Acc | H-score | OSCR | 50% sym Acc | H-score | OSCR | 80% sym Acc | H-score | OSCR | 50% asym Acc | H-score | OSCR |
|---|---|---|---|---|---|---|---|---|---|---|---|---|
| NPN Sheng et al. (2024) | 38.49 | 23.03 | 29.43 | 42.66 | 21.91 | 28.83 | 17.43 | 8.02 | 9.33 | 58.39 | 29.98 | 42.38 |
| BadLabel Zhang et al. (2024) | 46.31 | 38.81 | 44.12 | 36.70 | 26.94 | 33.37 | 17.44 | 5.40 | 8.30 | 38.05 | 33.01 | 33.26 |
| ODGNet Bose et al. (2023) | 68.88 | 40.87 | 51.07 | 60.90 | 30.62 | 43.12 | 17.17 | 11.31 | 9.52 | 45.96 | 26.46 | 32.41 |
| MLDG Chen et al. (2022) | 65.66 | 30.50 | 49.93 | 44.95 | 33.85 | 32.31 | 17.23 | 5.91 | 9.00 | 57.19 | 29.38 | 41.57 |
| MEDIC-cls Wang et al. (2023) | 20.09 | 12.23 | 8.94 | 17.17 | 10.21 | 5.51 | 18.21 | 7.34 | 8.83 | 16.44 | 11.10 | 7.37 |
| MEDIC-bcls Wang et al. (2023) | 20.09 | 13.65 | 6.76 | 17.17 | 12.80 | 5.28 | 17.73 | 7.47 | 8.30 | 16.44 | 14.55 | 6.98 |
| EBiL-HaDS-cls Peng et al. (2024b) | 63.96 | 43.81 | 49.39 | 53.93 | 32.34 | 39.33 | 15.56 | 9.36 | 7.45 | 56.07 | 35.75 | 38.38 |
| EBiL-HaDS-bcls Peng et al. (2024b) | 63.96 | 45.64 | 45.36 | 53.93 | 31.73 | 28.42 | 15.56 | 9.52 | 7.31 | 56.07 | 38.18 | 41.37 |
| HyProMeta Peng et al. (2024a) | 72.00 | 43.28 | 55.34 | 61.44 | 35.79 | 44.10 | 20.32 | 17.97 | 10.63 | 65.26 | 42.54 | 48.79 |
| Ours | 76.60 | 52.84 | 61.34 | 62.78 | 44.94 | 47.19 | 20.76 | 18.12 | 11.57 | 67.54 | 41.97 | 50.97 |

Table 6: Results (%) of DigitsDG on ConvNet. The open-set ratio is 6:4. The average domain performance is reported.

leave-one-domain-out setting Wang et al. (2023), using OSCR as the primary metric, with H-score and Acc as secondary metrics.

### 4.3 IMPLEMENTATION DETAILS

The experiments are all conducted by PyTorch2.0 on one NVIDIA A100 GPU. Training is limited to $1 \times 10^4$ steps, utilizing the SGD optimizer with a learning rate (LR) of $1 \times 10^{-3}$ and a batch size of 16. A learning rate decay of $1 \times 10^{-1}$ is applied after $8 \times 10^3$ meta-training steps. During the residual flow matching training, DiT Peebles & Xie (2023) is utilized as the backbone, where the training batch size is set as 128. $N_e$ is chosen as 10. Regarding the feature learning backbones, the ConvNet Zhou et al. (2021) is employed as the backbone network on the DigitsDG dataset, following Zhou et al. (2021). EReLiFM is only applied during training. In inference, no DiT structure is required, and the prediction relies solely on the chosen backbone and a lightweight classification head. This ensures test-time efficiency. For reference, the backbones used have parameter counts of $\sim 11.7M$ (ResNet18), $\sim 86M$ (ViT-Base), and $\sim 1.4M$ (ConvNet). $\lambda_c$, $\lambda_{dr}$, and $\lambda_{cr}$ are chosen as 1, 0.1, and 0.1, while $\lambda_p$ and $\lambda_{nc}$ are chosen as 1 and 1 equally according to the performance on the validation set.

### 4.4 COMPARISON BASELINES

For fair evaluation under the OSDG-NL setting, we compare against baselines that are compatible with domain generalization methods. ARPL, ODGNet, MLDG, SWAD, MixStyle, MEDIC, and EBiL-HaDS are established open-set domain generalization methods, while MLDG, MEDIC, EBiL-HaDS, and HyProMeta also serve as meta-learning approaches from the open-set domain generalization field. HyProMeta is the only existing method specifically designed for OSDG-NL and thus provides the most directly comparable baseline. Methods designed for closed-set noisy-label learning (*e.g.*, TCL, NPN, BadLabel) are also included, not as direct OSDG baselines, but to provide additional evaluation from a label-noise learning perspective. This ensures that our comparisons cover OSDG, noisy-label learning, and OSDG-NL dimensions.

## 4.5 ANALYSIS OF THE MODEL PERFORMANCE

In Tab. 1, Tab. 2, Tab. 3, and Tab. 4, we present performance comparisons between our proposed approach and other related methods. Among these, TCL Huang et al. (2023), NPN Sheng et al. (2024), BadLabel Zhang et al. (2024), DISC Li et al. (2023), LSL Kim et al. (2024), and PLM Zhao et al. (2024) focus on label noise learning, while MEDIC Wang et al. (2023), MLDG Shu et al. (2019), ARPL Chen et al. (2022), MixStyle Zhou et al. (2020c), ODGNet Bose et al. (2023), SWAD Cha et al. (2021), and EBiL-HaDS Peng et al. (2024b) specifically target open-set domain generalization. HyProMeta Peng et al. (2024a) is the first work addressing the OSDG-NL problem, utilizing hyperbolic prototypes to guide meta-learning. Although HyProMeta achieves the best performance among existing baselines, its reliance on a limited number of label-clean samples from the source domains and known classes constrains the model's generalization capability for OSDG-NL.

Compared to HyProMeta Peng et al. (2024a), our approach achieves 14.76%, 11.56%, 5.14%, and 11.99% accuracy improvements, 5.63%, 4.68%, 9.14%, and 3.63% H-score improvements, and 13.70%, 9.01%, 4.93%, and 8.20% OSCR improvements on the PACS dataset Li et al. (2017) using ResNet18 He et al. (2016) as the feature learning backbone, under symmetric label noise ratios of 20%, 50%, 80%, and asymmetric label noise ratio 50%, respectively. These improvements stem from residual flow matching, which enriches cross-category/domain paths, and UTS-ELC, which reliably separates clean from noisy labels. This allows effective optimization on limited clean data, while evidential learning further extracts cues from noisy samples during meta-test. We also find larger gains on visually rich domains (*i.e.*, *photo*, *art painting*, *cartoon*) than on *sketch*; under 80% symmetric noise, OSCR improvement on *sketch* is only 1.35%, indicating our method is most effective when visual features are preserved. EReLiFM outperforms HyProMeta because it addresses the weaknesses of prototype-based alignment at multiple levels. First, evidential training dynamics clustering separates clean from noisy samples, ensuring that training is guided by reliability-aware representations rather than corrupted prototypes. Second, domain- and category-conditioned residual flow matching models the distributional transport across domains and categories, capturing richer variations than simple mean-level alignment. Finally, the proposed evidential reliability-aware residual flow meta-learning pipeline systematically leverages clean, augmented, and cautiously recycled noisy data to expand the range of training tasks, thereby narrowing the gap to unseen domains. Together, these components form a principled framework that is theoretically more robust than HyProMeta, which relies solely on prototype matching.

## 4.6 CROSS-BACKBONE GENERALIZABILITY

To assess the cross-backbone generalizability, we conduct experiments using the ViT-Base Dosovitskiy et al. (2021) backbone on PACS Li et al. (2017) under the four label noise settings, as presented in Tab. 5. We first observe that employing a larger transformer architecture leads to overall performance improvements across all methods. Notably, HyProMeta Peng et al. (2024a) achieves 6.77%, 8.93%, 0.17%, and 5.64% OSCR improvements when using ViT-Base compared to ResNet18 He et al. (2016). Similar trends are observed in the performance of our proposed approach. Compared to the current state-of-the-art method, *i.e.*, HyProMeta Peng et al. (2024a), our approach achieves 13.42%, 13.12%, 4.57%, and 5.06% accuracy improvements, 0.94%, 1.46%, 7.66%, and 7.14% H-score improvements, and 6.27%, 10.30%, 9.70%, and 7.16% OSCR improvements under symmetric label noise ratios of 20%, 50%, 80%, and asymmetric label noise ratio of 50%, respectively. Per-target domain results are reported in the appendix.

## 4.7 EVALUATION ON ANOTHER DATASET

We further evaluate the generalizability of our proposed approach on the DigitsDG dataset, with results presented in Tab. 6. Several state-of-the-art methods with strong OSDG-NL performance are reported, including NPN Sheng et al. (2024), BadLabel Zhang et al. (2024), ODGNet Bose et al. (2023), MLDG Shu et al. (2019), MEDIC Wang et al. (2023), EBiL-HaDS Peng et al. (2024b), and HyProMeta Peng et al. (2024a). Among these, HyProMeta achieves the highest OSCR, with 55.34%, 44.10%, 10.63%, and 48.79% under symmetric label noise ratios of 20%, 50%, 80%, and asymmetric label noise ratio of 50%, respectively. Our approach consistently outperforms HyProMeta Peng et al. (2024a), achieving 61.34%, 47.19%, 11.57%, and 50.97% OSCR under the same noise settings. This improvement highlights the robustness of our method in handling noisy labels while ensuring effective

generalization across domains. Our approach benefits from residual flow matching, which enriches domain and category knowledge, and UTS-ELC, which improves clean-noisy label separation for robust meta-learning. Results confirm effectiveness across OSDG-NL datasets, including TerraInc (Tab. 16), where our method outperforms HyProMeta. Unlike prototype-based Peng et al. (2024a) or interpolation-based Zhou et al. (2020c) methods, which assume clean feature geometry or linear transition paths, residual flows approximate probabilistic transport maps between distributions. This theoretically provides a richer and more faithful modeling of domain- and category-conditioned shifts, and enables a more generalizable model optimization, especially when combined with evidential uncertainty for reliability-aware supervision during the label-noise-aware meta-learning stage. Further details can be found in the appendix.

## 4.8 ANALYSIS OF THE MODULE ABLATION

**Ablation of the DC-CRFM.** The ablation results are shown in Tab. 7. To evaluate the impact of DC-CRFM, we examine five model variants: *w/o DC-CRFM*, *w/o domain RA*, *w/o category RA*, *w/ mixup (replace DC-CRFM)*, and *w/ DirectFM*.

*w/o DC-CRFM* removes residual flow matching from meta-learning, *w/o domain RA* excludes augmentation of generated domain residuals, *w/o category RA* omits category residual augmentation, and *w/ mixup (replace DC-CRFM)* uses direct cross-domain and -class MixUp to replace DC-CRFM. Our results show that *w/o DC-CRFM* leads to $10.97\%$ and $6.45\%$ OSCR drop on target domains *mnist* and *syn*, highlighting the significance of using our proposed category and domain-conditioned residual flow matching in meta-learning. Additionally,

| Variants | mnist | | | syn | | |
|---|---|---|---|---|---|---|
| | ACC | H-score | OSCR | ACC | H-score | OSCR |
| w/o DC-CRFM | 71.89 | 17.80 | 58.91 | 50.19 | 39.04 | 33.19 |
| w/o domain RA | 76.03 | 30.23 | 66.66 | 50.17 | 35.59 | 30.20 |
| w/o category RA | 73.17 | 28.87 | 66.02 | 53.42 | 37.54 | 33.27 |
| w/ mixup (replace DC-CRFM) | 80.61 | 14.96 | 67.46 | 37.44 | 2.14 | 22.06 |
| w/ DirectFM | 75.44 | 62.33 | 63.28 | 54.69 | 34.56 | 37.08 |
| w/o UTS-ELC in RFM | 77.42 | 16.84 | 64.48 | 47.31 | 19.08 | 29.26 |
| w/ UTS-LC in RFM | 78.92 | 23.89 | 59.62 | 39.19 | 24.30 | 25.50 |
| w/o $\mathcal{L}_{EL}$ in meta-test | 78.42 | 23.33 | 61.15 | 52.58 | 36.62 | 33.68 |
| w/o $\mathcal{L}_{CE}$ in meta-test | 69.89 | 60.09 | 56.24 | 38.17 | 16.10 | 24.36 |
| **Ours** | **85.97** | **64.79** | **69.88** | **56.61** | **41.60** | **39.64** |

Table 7: Module ablation on the DigitsDG dataset, symmetric label noise with ratio $50\%$ is selected.

our approach consistently outperforms *w/o domain RA*, *w/o category RA*, and *w/ mixup (replace DC-CRFM)*, demonstrating the superior design of DC-CRFM for the OSDG-NL task. *w/ DirectFM* indicates that we do not learn residuals but use flow matching to generate images as augmentation. Notably, DC-CRFM consistently outperforms MixUp (*w/ mixup (replace DC-CRFM)*) by large margins, confirming that flow matching is not a simple interpolation-based augmentation. Instead, it learns structured residuals conditioned on domains and categories, enabling richer transferable paths.

**Ablation of the clean/noisy dataset partition technique.** We present two variants: *w/o UTS-ELC in RFM* and *w/ UTS-LC in RFM*. In the *w/o UTS-ELC in RFM* setting, residual flow matching is trained on the entire dataset without performing any clean/noisy separation; that is, all samples (clean and noisy) are treated uniformly when learning residuals. In the *w/ UTS-LC in RFM*, we still perform clean/noisy clustering using standard loss trajectories (UTS-LC), but the evidential learning loss is removed—meaning that the clustering relies only on plain cross-entropy trajectories, without uncertainty modeling. This ablation isolates the impact of evidential loss on clean/noisy partition quality and shows how our method behaves when the clustering signal becomes less reliable. Our approach outperforms both variants by $> 5\%$ OSCR, demonstrating the importance of proper label-clean/noisy data partitioning and the benefit of using evidential learning loss.

**Ablation of meta-learning task.** We further conduct another ablation regarding the meta-learning by removing the evidential pseudo-label supervision in the meta-test stage, indicated by *w/o $\mathcal{L}_{EL}$*. Our proposed method contributes $8.73\%$ and $5.96\%$ performance gains in terms of OSCR, illustrating the superiority of using evidential pseudo-label supervision on the label-noisy set during the meta-training for the model optimization. On the other hand, the variant *w/o $\mathcal{L}_{CE}$* shows a performance drop, indicating the importance of both losses. While $\mathcal{L}_{EL}$ enables label correction, $\mathcal{L}_{CE}$ helps extract useful cues from misassigned clean samples in the noisy set.

## 5 CONCLUSION

We present **EReLiFM**, a reliability-aware residual flow meta-learning framework for open-set domain generalization under noisy labels. By combining evidential clustering for clean/noisy data separation with domain- and category-conditioned flow matching, our method enhances data reliability and diversity for meta-learning. Experiments on multiple benchmarks confirm that EReLiFM achieves robust performance against label noise and strong generalization to unseen domains and categories.

## REPRODUCIBILITY STATEMENT

The source code of our proposed approach is available at `https://anonymous.4open.science/r/ERELIFM-CBCB` to ensure reproducibility.

## ETHICS STATEMENT

This work presents a methodological contribution to open-set domain generalization under noisy labels and is conducted entirely on publicly available benchmark datasets that do not involve human subjects or sensitive personal information. The research does not raise concerns regarding privacy, security, fairness, bias, or potential harmful applications, and complies with accepted standards of research integrity and ethical practice.

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

APPENDIX

## A  THE USE OF LARGE LANGUAGE MODELS (LLMs)

In this work, we mainly rely on LLM for text rephrasing to polish the paper writing.

## B  SOCIAL IMPACT AND LIMITATIONS

**Social impact:** The proposed EReLiFM framework has a significant social impact by improving model generalization to new categories and domains under noisy labels, which is critical for real-world applications such as healthcare, security, and autonomous driving. By facilitating robust learning in open-set environments, this work enhances the reliability of deep learning models deployed in dynamic and uncertain conditions when label noise exists. The ability to manage label noise ensures that models trained on imperfect annotations, such as crowdsourced data, maintain their effectiveness and trustworthiness. Furthermore, our approach mitigates biases in deep learning-based decision-making by distinguishing between reliable and noisy labels, contributing to fairer and more accountable deep learning systems. However, the potential misclassification and biased prediction remain, which could lead to erroneous decisions with adverse societal implications.

**Limitations:** We propose EReLiFM to mitigate label noise in OSDG, but its performance under extreme noise remains limited, highlighting a key research direction. This work focuses on image-based OSDG-NL, leaving video-based OSDG-NL for future exploration.

## C  MORE DETAILS REGARDING THE EVALUATION METRICS

We follow the protocol outlined in the MEDIC approach Wang et al. (2023). For the PACS Li et al. (2017) dataset, we adopt an open-set ratio of $6:1$, designating *elephant*, *horse*, *giraffe*, *dog*, *guitar*, and *house* as seen categories, while *person* is treated as unseen. Similarly, in DigitsDG Zhou et al. (2020a), we use an open-set ratio of $6:4$, with digits $0,1,2,3,4,5$ as seen and $6,7,8,9$ as unseen.

For evaluation, we employ three metrics. *Acc* measures closed-set accuracy on seen categories, while *H-score* and *OSCR* assess open-set recognition. The *H-score*, dependent on a threshold from the source domain validation set, is considered as the secondary metric. In contrast, OSCR, introduced by MEDIC Wang et al. (2023), evaluates open-set recognition without a predefined threshold, making it our primary metric.

The *H-score* is computed using a threshold ratio $\lambda$ to distinguish seen from unseen samples. Predictions below $\lambda$ are classified as unseen, and accuracy is separately calculated for seen ($Acc_k$) and unseen ($Acc_u$) categories. The final *H-score* is given by:

$$H_{score} = \frac{2 \times Acc_u \times Acc_k}{Acc_u + Acc_k}. \tag{7}$$

OSCR, unlike AUROC, integrates accuracy with AUROC through dynamic thresholding, focusing only on correctly classified samples. It combines elements from both *H-score* and AUROC, offering a more comprehensive measure of confidence reliability in OSDG tasks.

## D  ANALYSIS OF CONFIDENCE SCORE

Fig. 2 visualizes confidence scores for seen (red) and unseen (blue) categories, computed as the maximum Softmax probability. Our approach achieves the best separation between seen categories and unseen categories on the test domain, while the confidence scores delivered by other listed baselines are merged together. This visualization illustrates the superior capability of the proposed approach when it deals with out-of-distribution categories. The proposed categorical flow matching improves the awareness of unseen categories during the representation learning.

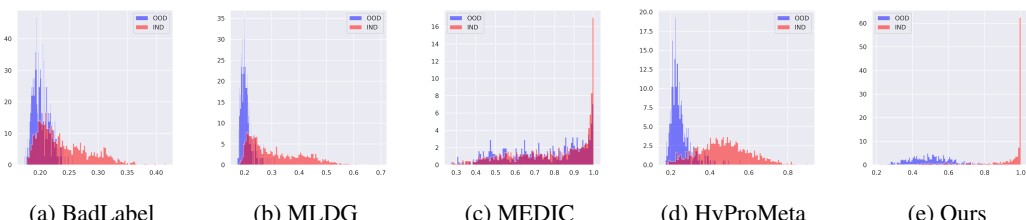

|          |          |          |          |          |
|----------|----------|----------|----------|----------|
| (a) BadLabel | (b) MLDG | (c) MEDIC | (d) HyProMeta | (e) Ours |

Figure 2: Confidence score visualization of learned representations on PACS with target domain *photo*, using ResNet18 He et al. (2016) under symmetric label noise with ratio $50\%$.

| Method | Photo (P) | | | Art (A) | | | Cartoon (C) | | | Sketch (S) | | | Avg | | |
|--------|-----|---------|------|-----|---------|------|-----|---------|------|-----|---------|------|-----|---------|------|
| | Acc | H-score | OSCR | Acc | H-score | OSCR | Acc | H-score | OSCR | Acc | H-score | OSCR | Acc | H-score | OSCR |
| TCL Huang et al. (2023) | 67.77 | 79.17 | 67.64 | 64.35 | 63.45 | 56.91 | 51.32 | 50.24 | 42.30 | 26.43 | 28.25 | 20.54 | 52.47 | 55.28 | 46.85 |
| NPN Sheng et al. (2024) | 63.97 | 71.61 | 62.53 | 57.22 | 51.43 | 45.95 | 47.65 | 40.11 | 32.85 | 21.86 | 13.01 | 12.71 | 47.68 | 44.04 | 38.51 |
| BadLabel Zhang et al. (2024) | 62.66 | 64.81 | 57.87 | 51.59 | 60.49 | 49.13 | 50.85 | 58.42 | 47.75 | 31.15 | 18.54 | 29.40 | 49.06 | 50.57 | 46.04 |
| DISC Li et al. (2023) | 66.69 | 71.98 | 65.67 | 54.10 | 29.80 | 41.07 | 53.48 | 41.72 | 40.83 | 34.57 | 16.94 | 24.16 | 52.21 | 40.11 | 42.93 |
| LSL Kim et al. (2024) | 67.04 | 71.46 | 63.82 | 63.10 | 63.42 | 58.01 | 53.27 | 54.92 | 47.14 | 28.44 | 4.37 | 27.51 | 52.96 | 48.54 | 49.12 |
| PLM Zhao et al. (2024) | 65.73 | 61.01 | 59.76 | 65.73 | 61.01 | 59.76 | 52.71 | 48.17 | 46.32 | 27.57 | 17.94 | 20.62 | 52.94 | 47.03 | 46.62 |
| ARPL Bendale & Boult (2016) | 68.09 | 75.04 | 67.31 | 56.91 | 51.96 | 44.50 | 59.93 | 60.98 | 54.02 | 36.46 | 7.36 | 27.55 | 55.35 | 48.84 | 48.35 |
| ODGNet Bose et al. (2023) | 66.48 | 71.17 | 65.60 | 62.44 | 65.75 | 58.60 | 60.50 | 59.99 | 53.54 | 30.14 | 4.05 | 15.68 | 54.89 | 50.24 | 48.36 |
| MLDG Shu et al. (2019) | 66.16 | 72.74 | 64.05 | 57.85 | 54.34 | 47.79 | 60.80 | 61.93 | 54.69 | 36.38 | 11.21 | 29.16 | 55.30 | 50.06 | 48.92 |
| SWAD Cha et al. (2021) | 63.00 | 72.01 | 62.08 | 64.79 | 66.60 | 60.22 | 56.68 | 59.18 | 51.10 | 29.90 | 20.51 | 22.51 | 53.59 | 54.58 | 48.98 |
| MixStyle Zhou et al. (2020c) | 68.42 | 63.75 | 59.88 | 63.23 | 61.42 | 56.49 | 51.99 | 54.34 | 45.19 | 28.36 | 3.13 | 12.79 | 53.00 | 45.66 | 43.59 |
| MEDIC-cls Wang et al. (2023) | 65.83 | 70.15 | 62.11 | 66.04 | 55.36 | 57.33 | 56.42 | 59.05 | 51.07 | 38.76 | 26.01 | 21.45 | 56.76 | 52.64 | 47.99 |
| MEDIC-bcls Wang et al. (2023) | 65.83 | 70.08 | 63.40 | 66.04 | 55.18 | 55.40 | 56.42 | 54.09 | 49.89 | 38.76 | 14.81 | 25.12 | 56.76 | 48.54 | 48.45 |
| EBiL-HaDS-cls Peng et al. (2024b) | 65.43 | 55.59 | 50.68 | 65.48 | 65.80 | 60.71 | 56.83 | 58.38 | 50.43 | 37.22 | 18.15 | 24.87 | 56.24 | 49.48 | 46.67 |
| EBiL-HaDS-bcls Peng et al. (2024b) | 65.43 | 53.39 | 55.16 | 65.48 | 59.68 | 58.33 | 56.83 | 56.59 | 48.76 | 37.22 | 19.38 | 23.75 | 56.24 | 47.26 | 46.50 |
| HyProMeta Peng et al. (2024a) | 68.90 | 80.47 | 68.86 | 68.17 | 70.60 | 63.65 | 61.47 | 62.10 | 55.15 | 40.06 | 27.06 | 32.21 | 59.65 | 60.06 | 54.97 |
| Ours | 94.26 | 80.16 | 81.49 | 86.12 | 72.33 | 80.79 | 72.67 | 64.00 | 61.67 | 39.24 | 27.50 | 21.01 | 73.07 | 61.00 | 61.24 |

Table 8: Results (%) of PACS Li et al. (2017) on ViT-Base Dosovitskiy et al. (2021). The open-set ratio is 6:1 and symmetric label noise with ratio $20\%$ is selected.

| Method | Photo (P) | | | Art (A) | | | Cartoon (C) | | | Sketch (S) | | | Avg | | |
|--------|-----|---------|------|-----|---------|------|-----|---------|------|-----|---------|------|-----|---------|------|
| | Acc | H-score | OSCR | Acc | H-score | OSCR | Acc | H-score | OSCR | Acc | H-score | OSCR | Acc | H-score | OSCR |
| TCL Huang et al. (2023) | 68.17 | 62.70 | 67.56 | 64.48 | 67.84 | 61.30 | 45.80 | 35.84 | 35.68 | 22.29 | 9.28 | 3.54 | 50.19 | 43.92 | 42.02 |
| NPN Sheng et al. (2024) | 27.71 | 10.79 | 8.48 | 39.34 | 33.88 | 32.40 | 40.39 | 44.98 | 34.61 | 20.80 | 26.62 | 19.84 | 32.06 | 29.07 | 23.83 |
| BadLabel Zhang et al. (2024) | 51.62 | 61.25 | 50.41 | 32.52 | 37.26 | 26.67 | 43.58 | 55.85 | 42.45 | 31.60 | 28.24 | 26.93 | 39.83 | 45.65 | 36.62 |
| DISC Li et al. (2023) | 57.35 | 52.08 | 44.73 | 38.02 | 37.36 | 28.09 | 30.84 | 31.69 | 22.68 | 20.72 | 16.32 | 16.20 | 36.73 | 34.36 | 27.93 |
| LSL Kim et al. (2024) | 65.99 | 68.68 | 62.18 | 55.66 | 49.85 | 44.97 | 49.20 | 44.94 | 36.86 | 29.90 | 6.91 | 19.18 | 50.19 | 42.60 | 40.80 |
| PLM Zhao et al. (2024) | 57.67 | 50.81 | 51.87 | 51.41 | 46.57 | 41.82 | 39.66 | 39.47 | 31.89 | 19.95 | 12.93 | 15.75 | 42.17 | 37.45 | 35.33 |
| ARPL Bendale & Boult (2016) | 57.27 | 62.95 | 53.89 | 39.21 | 37.92 | 29.78 | 61.62 | 53.40 | 45.17 | 33.03 | 2.77 | 17.23 | 45.28 | 39.26 | 36.52 |
| ODGNet Bose et al. (2023) | 68.09 | 76.73 | 67.32 | 64.79 | 64.09 | 59.64 | 53.22 | 52.74 | 47.68 | 34.39 | 20.16 | 18.61 | 55.12 | 53.43 | 48.31 |
| MLDG Shu et al. (2019) | 67.29 | 75.04 | 65.97 | 66.98 | 66.41 | 62.00 | 55.44 | 54.65 | 47.35 | 33.83 | 15.82 | 23.25 | 55.89 | 52.98 | 49.64 |
| SWAD Cha et al. (2021) | 68.58 | 78.86 | 68.33 | 63.29 | 65.39 | 58.28 | 51.68 | 52.20 | 44.65 | 26.48 | 24.19 | 16.84 | 52.51 | 55.16 | 47.03 |
| MixStyle Zhou et al. (2020c) | 54.04 | 53.94 | 46.23 | 52.72 | 46.49 | 39.10 | 37.65 | 29.78 | 22.47 | 20.38 | 2.35 | 13.29 | 41.20 | 33.14 | 30.27 |
| MEDIC-cls Wang et al. (2023) | 62.76 | 69.29 | 60.03 | 63.10 | 62.25 | 55.06 | 56.42 | 56.30 | 49.08 | 32.16 | 8.13 | 24.16 | 53.61 | 48.99 | 47.08 |
| MEDIC-bcls Wang et al. (2023) | 62.76 | 50.05 | 44.19 | 63.10 | 45.67 | 46.56 | 56.42 | 50.59 | 40.02 | 32.16 | 27.74 | 23.91 | 53.61 | 40.86 | 38.67 |
| EBiL-HaDS-cls Peng et al. (2024b) | 62.84 | 64.75 | 57.55 | 63.79 | 62.07 | 56.86 | 48.07 | 45.94 | 38.18 | 36.00 | 10.31 | 24.98 | 52.68 | 45.77 | 44.39 |
| EBiL-HaDS-bcls Peng et al. (2024b) | 62.84 | 45.39 | 40.81 | 63.79 | 56.27 | 51.76 | 48.07 | 43.14 | 34.07 | 36.00 | 10.83 | 16.06 | 52.68 | 38.91 | 35.68 |
| HyProMeta Peng et al. (2024a) | 68.80 | 80.58 | 68.72 | 67.60 | 68.10 | 62.07 | 58.95 | 58.32 | 51.68 | 39.37 | 30.30 | 29.16 | 58.68 | 59.33 | 52.91 |
| Ours | 83.20 | 78.55 | 77.30 | 87.43 | 79.11 | 82.34 | 63.43 | 58.69 | 53.91 | 53.12 | 26.82 | 39.28 | 71.80 | 60.79 | 63.21 |

Table 9: Results (%) of PACS on ViT-Base Dosovitskiy et al. (2021). The open-set ratio is 6:1 and symmetric label noise with ratio $50\%$ is selected.

# E  PER-TARGET-DOMAIN RESULTS ON PACS USING VIT-BASE AND DIGITSDG USING CONVNET

We further deliver the per-target-domain performances for the experiments conducted on the PACS Li et al. (2017) dataset using ViT-Base Dosovitskiy et al. (2021) backbone (as shown in Tab. 8, Tab. 9, Tab. 10, and Tab. 11), and the experiments conducted on the DigitsDG Zhou et al. (2020a) dataset using ConvNet Zhou et al. (2021) backbone (as shown in Tab. 12, Tab. 13, Tab. 14, and Tab. 15). From the aforementioned tables, we can observe that our proposed approach consistently outperforms the others across all the metrics and label noise settings in general, which demonstrates the superior generalizability of our approach across different backbones, label noise settings, and datasets.

| Method | Photo (P) | | | Art (A) | | | Cartoon (C) | | | Sketch (S) | | | Avg | | |
|---|---|---|---|---|---|---|---|---|---|---|---|---|---|---|---|
| | Acc | H-score | OSCR | Acc | H-score | OSCR | Acc | H-score | OSCR | Acc | H-score | OSCR | Acc | H-score | OSCR |
| TCL Huang et al. (2023) | 22.70 | 25.47 | 16.55 | 22.26 | 23.14 | 14.42 | 19.75 | 14.23 | 6.53 | 28.52 | 10.92 | 11.54 | 23.31 | 18.44 | 12.26 |
| NPN Sheng et al. (2024) | 15.27 | 15.29 | 7.9 | 16.07 | 10.88 | 11.29 | 19.96 | 17.19 | 8.06 | 20.48 | 1.12 | 18.43 | 17.95 | 11.12 | 11.42 |
| BadLabel Zhang et al. (2024) | 13.33 | 20.96 | 13.32 | 24.77 | 32.36 | 21.98 | 23.98 | 32.76 | 21.91 | 21.60 | 2.36 | **21.31** | 20.92 | 22.11 | 19.63 |
| DISC Li et al. (2023) | 22.37 | 1.92 | 11.07 | 24.58 | 19.96 | 12.81 | 24.08 | 13.40 | 14.20 | 20.03 | 4.56 | 12.61 | 22.77 | 9.89 | 12.67 |
| LSL Kim et al. (2024) | 10.02 | 3.64 | 5.89 | 26.27 | 10.83 | 17.29 | 31.87 | 27.63 | 19.75 | 25.39 | 8.09 | 7.83 | 23.39 | 12.55 | 12.69 |
| PLM Zhao et al. (2024) | 29.32 | 5.93 | 1.34 | 26.77 | 28.58 | 18.39 | 25.27 | 25.69 | 16.89 | 24.22 | 3.33 | 2.25 | 26.40 | 15.88 | 9.72 |
| ARPL Bendale & Boult (2016) | 25.85 | 20.35 | 21.54 | 20.14 | 18.60 | 13.63 | 24.24 | 23.62 | 14.23 | 16.26 | 3.36 | 9.32 | 21.62 | 16.48 | 14.68 |
| ODGNet Bose et al. (2023) | 23.18 | 19.01 | 10.04 | 22.64 | 21.69 | 10.84 | 19.80 | 19.01 | 10.58 | 18.07 | 0.60 | 1.60 | 20.92 | 15.08 | 8.27 |
| MLDG Shu et al. (2019) | 29.08 | 29.84 | 22.82 | 30.08 | 29.62 | 20.16 | 24.24 | 22.37 | 15.85 | 19.66 | 2.34 | 4.11 | 25.77 | 21.04 | 15.74 |
| SWAD Cha et al. (2021) | 19.42 | 15.01 | 15.83 | 23.83 | 22.54 | 14.46 | 28.14 | 26.18 | 16.29 | 23.59 | 14.73 | 9.01 | 23.75 | 19.62 | 13.90 |
| MixStyle Zhou et al. (2020c) | 22.70 | 25.94 | 18.23 | 17.32 | 8.52 | 15.69 | 34.97 | 24.21 | 16.86 | 20.99 | 1.21 | 10.62 | 24.00 | 14.97 | 15.35 |
| MEDIC-cls Wang et al. (2023) | 35.70 | 23.85 | 14.91 | 28.46 | 30.78 | 20.50 | 30.22 | 26.39 | 16.88 | 21.51 | 11.08 | 13.13 | 28.97 | 23.03 | 16.36 |
| MEDIC-bcls Wang et al. (2023) | 35.70 | 23.85 | 18.53 | 28.46 | 27.55 | 17.87 | 30.22 | 13.99 | 11.18 | 21.54 | 8.32 | 4.81 | 28.98 | 18.43 | 13.10 |
| EBiL-HaDS-cls Peng et al. (2024b) | 34.89 | 3.49 | 17.80 | 32.02 | 20.65 | 22.32 | 30.84 | 6.28 | 17.64 | 25.71 | 26.60 | 17.09 | 30.87 | 14.26 | 18.71 |
| EBiL-HaDS-bcls Peng et al. (2024b) | 34.89 | 20.63 | 16.79 | 32.02 | 26.79 | 17.85 | 30.84 | 24.30 | 16.61 | 25.71 | **29.67** | 16.33 | 30.87 | 25.35 | 16.90 |
| HyProMeta Peng et al. (2024a) | 41.03 | 29.98 | 26.92 | 39.84 | 39.82 | 30.06 | 36.31 | 33.05 | 23.79 | **31.07** | 13.50 | 20.26 | 37.06 | 29.09 | 25.26 |
| Ours | **44.59** | **32.94** | **54.15** | **57.63** | **48.27** | **35.88** | **42.91** | **37.99** | **31.17** | 21.40 | 27.80 | 18.63 | **41.63** | **36.75** | **34.96** |

Table 10: Results (%) of PACS on ViT-Base Dosovitskiy et al. (2021). The open-set ratio is 6:1 and symmetric label noise with ratio $80\%$ is selected.

| Method | Photo (P) | | | Art (A) | | | Cartoon (C) | | | Sketch (S) | | | Avg | | |
|---|---|---|---|---|---|---|---|---|---|---|---|---|---|---|---|
| | Acc | H-score | OSCR | Acc | H-score | OSCR | Acc | H-score | OSCR | Acc | H-score | OSCR | Acc | H-score | OSCR |
| TCL Huang et al. (2023) | 51.21 | 56.51 | 48.47 | 46.65 | 51.35 | 40.80 | 31.92 | 30.94 | 22.61 | 25.87 | 26.99 | 12.04 | 38.91 | 39.94 | 30.98 |
| NPN Sheng et al. (2024) | 27.63 | 8.86 | 6.32 | 32.40 | 19.00 | 15.72 | 21.66 | 16.87 | 14.25 | 20.48 | 26.15 | 16.50 | 25.54 | 17.72 | 13.20 |
| BadLabel Zhang et al. (2024) | 46.12 | 58.96 | 45.77 | 35.33 | 34.30 | 34.83 | 26.92 | 39.52 | 26.18 | 20.70 | **31.32** | 20.66 | 32.27 | 41.03 | 31.86 |
| DISC Li et al. (2023) | 47.17 | 17.08 | 9.77 | 24.64 | 19.06 | 10.75 | 20.22 | 12.84 | 9.22 | 23.93 | 4.31 | 15.10 | 28.99 | 13.55 | 11.21 |
| LSL Kim et al. (2024) | 49.52 | 5.63 | 15.99 | 37.52 | 34.68 | 26.68 | 28.16 | 27.60 | 18.13 | 28.16 | 27.60 | 18.13 | 35.84 | 23.88 | 19.73 |
| PLM Zhao et al. (2024) | 20.51 | 23.01 | 18.15 | 20.17 | 13.41 | 8.08 | 37.85 | 33.88 | 26.43 | 20.59 | 24.16 | 17.26 | 24.78 | 23.62 | 17.48 |
| ARPL Bendale & Boult (2016) | 50.32 | 53.47 | 44.10 | 44.03 | 42.48 | 33.24 | 43.15 | 36.75 | 30.20 | 17.72 | 13.62 | 12.77 | 38.81 | 36.58 | 30.08 |
| ODGNet Bose et al. (2023) | 51.13 | 55.83 | 48.95 | 49.16 | 45.58 | 37.21 | 40.69 | 37.35 | 28.85 | 21.76 | 17.55 | 12.68 | 40.69 | 39.08 | 31.92 |
| MLDG Shu et al. (2019) | 52.99 | 57.10 | 51.53 | 45.90 | 56.06 | 43.93 | 46.11 | 43.18 | 39.17 | 32.98 | 29.49 | 28.18 | 44.50 | 46.46 | 40.70 |
| SWAD Cha et al. (2021) | 50.32 | 54.99 | 45.45 | 44.84 | 54.03 | 42.11 | 39.92 | 45.57 | 35.45 | 25.42 | 18.78 | 12.07 | 40.13 | 43.34 | 33.77 |
| MixStyle Zhou et al. (2020c) | 53.72 | 53.47 | 52.97 | 46.15 | 48.22 | 39.13 | 44.87 | 44.84 | 36.37 | 29.50 | 12.81 | 19.03 | 43.56 | 39.84 | 36.88 |
| MEDIC-cls Wang et al. (2023) | 52.10 | 59.31 | 48.42 | 46.47 | 56.57 | 44.42 | 35.69 | 29.04 | 21.31 | 35.42 | 28.01 | 27.79 | 42.42 | 43.23 | 35.49 |
| MEDIC-bcls Wang et al. (2023) | 52.10 | 49.72 | 42.70 | 46.47 | 55.52 | 43.88 | 35.69 | 30.26 | 20.16 | 35.42 | 24.54 | 26.48 | 42.42 | 40.01 | 33.31 |
| EBiL-HaDS-cls Peng et al. (2024b) | 54.60 | 52.92 | 44.98 | 46.97 | 56.52 | 44.29 | 34.66 | 25.52 | 22.48 | 25.71 | 26.60 | 17.09 | 40.49 | 40.39 | 32.21 |
| EBiL-HaDS-bcls Peng et al. (2024b) | 54.60 | 39.01 | 29.61 | 46.97 | 54.34 | 42.74 | 34.66 | 31.86 | 22.47 | 25.71 | 29.67 | 16.33 | 40.49 | 38.72 | 27.79 |
| HyProMeta Peng et al. (2024a) | 56.87 | 59.59 | 53.15 | 55.97 | 56.31 | 48.31 | 48.94 | 46.85 | 40.16 | **38.18** | 31.14 | **32.15** | 49.99 | 48.47 | 43.44 |
| Ours | **85.06** | **80.45** | **83.81** | **62.41** | **59.56** | **55.45** | **49.25** | **51.11** | **42.75** | 23.48 | 31.30 | 20.37 | **55.05** | **55.61** | **50.60** |

Table 11: Results (%) of PACS on ViT-Base Dosovitskiy et al. (2021). The open-set ratio is 6:1 and asymmetric label noise with ratio $50\%$ is selected.

| Method | mnist | | | $mnist_m$ | | | syn | | | svhn | | | Avg | | |
|---|---|---|---|---|---|---|---|---|---|---|---|---|---|---|---|
| | Acc | H-score | OSCR | Acc | H-score | OSCR | Acc | H-score | OSCR | Acc | H-score | OSCR | Acc | H-score | OSCR |
| NPN Sheng et al. (2024) | 82.28 | 29.42 | 70.68 | 32.56 | 28.22 | 22.74 | 21.78 | 22.54 | 14.41 | 17.33 | 11.95 | 9.89 | 38.49 | 23.03 | 29.43 |
| BadLabel Zhang et al. (2024) | 63.25 | 52.49 | 61.18 | 30.00 | 34.82 | 27.42 | 50.17 | 51.19 | 46.06 | 41.82 | 16.75 | 41.82 | 46.31 | 38.81 | 44.12 |
| ODGNet Bose et al. (2023) | 90.33 | 50.84 | 71.38 | 59.28 | 26.10 | 43.83 | 70.11 | 53.83 | 49.54 | 55.81 | 32.71 | 39.54 | 68.88 | 40.87 | 51.07 |
| MLDG Chen et al. (2022) | 90.67 | 27.60 | 80.46 | 57.89 | 48.38 | 42.77 | 60.33 | 39.95 | 41.30 | 53.75 | 6.08 | 35.20 | 65.66 | 30.50 | 49.93 |
| MEDIC-cls Wang et al. (2023) | 22.08 | 8.71 | 5.89 | 21.33 | 20.09 | 9.94 | 23.24 | 9.83 | 13.28 | 13.72 | 10.29 | 6.63 | 20.09 | 12.23 | 8.94 |
| MEDIC-bcls Wang et al. (2023) | 22.08 | 12.31 | 5.57 | 21.33 | 16.19 | 10.11 | 23.24 | 12.64 | 5.41 | 13.72 | 6.43 | 5.94 | 20.09 | 13.65 | 6.76 |
| EBiL-HaDS-cls Peng et al. (2024b) | 88.28 | 48.19 | 78.11 | 42.86 | 33.81 | 29.72 | 72.36 | 54.16 | 51.49 | 52.33 | 39.08 | 38.24 | 63.96 | 43.81 | 49.39 |
| EBiL-HaDS-bcls Peng et al. (2024b) | 88.28 | 59.51 | 61.82 | 42.86 | 34.27 | 30.03 | 72.36 | 53.55 | 49.29 | 52.33 | 35.23 | 40.31 | 63.96 | 45.64 | 45.36 |
| HyProMeta Peng et al. (2024a) | 93.47 | 55.15 | 82.35 | 61.69 | 41.43 | 43.40 | 74.02 | 53.88 | 53.10 | 58.83 | 22.64 | 42.52 | 72.00 | 43.28 | 55.34 |
| Ours | **95.19** | **57.92** | **86.41** | **67.67** | **50.97** | **51.76** | **78.44** | **60.49** | **58.85** | **65.08** | **41.98** | **48.32** | **76.60** | **52.84** | **61.34** |

Table 12: Results (%) of DigitsDG on ConvNet Zhou et al. (2021), where symmetric label noise with ratio $20\%$ is selected.

| Method | mnist | | | $mnist_m$ | | | syn | | | svhn | | | Avg | | |
|---|---|---|---|---|---|---|---|---|---|---|---|---|---|---|---|
| | Acc | H-score | OSCR | Acc | H-score | OSCR | Acc | H-score | OSCR | Acc | H-score | OSCR | Acc | H-score | OSCR |
| NPN Sheng et al. (2024) | 68.11 | 25.80 | 49.14 | 28.31 | 28.29 | 18.65 | 45.78 | 32.17 | 31.15 | 28.42 | 1.38 | 16.37 | 42.66 | 21.91 | 28.83 |
| BadLabel Zhang et al. (2024) | 63.31 | 35.12 | 57.47 | 42.28 | 42.69 | 38.44 | 21.36 | 25.07 | 18.30 | 19.83 | 4.88 | 19.28 | 36.70 | 26.94 | 33.37 |
| ODGNet Bose et al. (2023) | 71.25 | 22.26 | 49.81 | 59.22 | 36.88 | 44.86 | **61.39** | **43.65** | **42.41** | **51.75** | 19.70 | **35.38** | 60.90 | 30.62 | 43.12 |
| MLDG Shu et al. (2019) | 62.72 | 54.21 | 50.52 | 48.94 | 44.54 | 35.35 | 43.53 | 33.15 | 29.09 | 24.61 | 6.48 | 14.29 | 44.95 | 33.85 | 32.31 |
| MEDIC-cls Wang et al. (2023) | 22.39 | 0.17 | 2.81 | 25.89 | 22.88 | 12.16 | 12.78 | 12.10 | 4.73 | 7.61 | 5.70 | 2.35 | 17.17 | 10.21 | 5.51 |
| MEDIC-bcls Wang et al. (2023) | 22.39 | 12.08 | 3.26 | 25.89 | 20.73 | 11.88 | 12.78 | 11.26 | 3.97 | 7.61 | 7.13 | 2.02 | 17.17 | 12.80 | 5.28 |
| EBiL-HaDS-cls Peng et al. (2024b) | 77.19 | 31.34 | 62.64 | 42.78 | 34.74 | 32.69 | 51.92 | 39.84 | 31.75 | 43.81 | 23.45 | 30.23 | 53.93 | 32.34 | 39.33 |
| EBiL-HaDS-bcls Peng et al. (2024b) | 77.19 | 47.43 | 47.86 | 42.78 | 30.28 | 22.24 | 51.92 | 18.79 | 22.75 | 43.81 | **30.42** | 20.81 | 53.93 | 31.73 | 28.42 |
| HyProMeta Peng et al. (2024a) | 82.39 | 31.35 | 63.53 | **60.41** | 46.23 | 46.37 | 55.33 | 41.29 | 34.72 | 47.64 | 24.29 | 31.79 | 61.44 | 35.79 | 44.10 |
| Ours | **85.97** | **64.79** | **69.88** | 60.31 | 49.11 | 47.34 | 56.61 | 41.60 | 39.64 | 48.22 | 24.26 | 31.89 | **62.78** | **44.94** | **47.19** |

Table 13: Results (%) of DigitsDG on ConvNet Zhou et al. (2021), where symmetric label noise with ratio $50\%$ is selected.

# F ABLATION OF UNSUPERVISED CLUSTERING FOR CLEAN-NOISY PARTITION

In Tab. 19, we present ablation experiments on the unsupervised clustering approach for label-clean/noisy set partitioning. Separation correctness is evaluated using accuracy, where a binary indicator serves as ground truth, denoting whether the current label matches the original unperturbed

| Method | mnist | | | mnist$_m$ | | | syn | | | svhn | | | Avg | | |
|---|---|---|---|---|---|---|---|---|---|---|---|---|---|---|---|
| | Acc | H-score | OSCR | Acc | H-score | OSCR | Acc | H-score | OSCR | Acc | H-score | OSCR | Acc | H-score | OSCR |
| NPN Sheng et al. (2024) | 16.67 | 0.01 | 9.65 | 18.61 | 13.38 | 10.15 | 17.78 | 18.12 | 9.29 | 16.67 | 0.55 | 8.21 | 17.43 | 8.02 | 9.33 |
| BadLabel Zhang et al. (2024) | 18.58 | 5.84 | 7.39 | 17.39 | 15.13 | 8.99 | 16.67 | 0.41 | 8.36 | 17.11 | 0.22 | 8.44 | 17.44 | 5.40 | 8.30 |
| ODGNet Bose et al. (2023) | 16.19 | 1.51 | 10.37 | 17.28 | 11.80 | 10.64 | 18.47 | 17.04 | 8.49 | 16.72 | 14.89 | 8.59 | 17.17 | 11.31 | 9.52 |
| MLDG Shu et al. (2019) | 16.06 | 6.70 | 9.69 | 18.58 | 3.26 | 9.27 | 16.94 | 6.81 | 8.20 | 17.33 | 6.88 | 8.84 | 17.23 | 5.91 | 9.00 |
| MEDIC-cls Wang et al. (2023) | 21.17 | 3.51 | 11.37 | 18.75 | 16.52 | 7.77 | 15.81 | 4.48 | 7.54 | 17.11 | 4.86 | 8.64 | 18.21 | 7.34 | 8.83 |
| MEDIC-bcls Wang et al. (2023) | 21.17 | 7.21 | 8.32 | 16.83 | 13.25 | 8.81 | 15.81 | 4.44 | 7.70 | 17.11 | 4.96 | 8.38 | 17.73 | 7.47 | 8.30 |
| EBiL-HaDS-cls Peng et al. (2024b) | 12.72 | 7.25 | 5.36 | 16.14 | 12.10 | 8.46 | 16.92 | 12.83 | 8.17 | 16.44 | 5.27 | 7.79 | 15.56 | 9.36 | 7.45 |
| EBiL-HaDS-bcls Peng et al. (2024b) | 12.72 | 7.15 | 5.62 | 16.14 | 8.64 | 7.43 | 16.92 | 15.29 | 8.34 | 16.44 | 7.00 | 7.83 | 15.56 | 9.52 | 7.31 |
| HyProMeta Peng et al. (2024a) | 22.28 | 20.94 | 12.23 | **21.58** | 16.92 | **11.73** | 19.31 | **18.47** | 9.77 | **18.11** | 15.55 | 8.78 | 20.32 | 17.97 | 10.63 |
| Ours | **25.33** | **23.82** | **14.14** | 18.64 | **20.03** | 10.85 | **21.89** | 12.67 | **12.02** | 17.17 | **15.94** | **9.26** | **20.76** | **18.12** | **11.57** |

Table 14: Results (%) of DigitsDG on ConvNet Zhou et al. (2021), where symmetric label noise with ratio 80% is selected.

| Method | mnist | | | mnist$_m$ | | | syn | | | svhn | | | Avg | | |
|---|---|---|---|---|---|---|---|---|---|---|---|---|---|---|---|
| | Acc | H-score | OSCR | Acc | H-score | OSCR | Acc | H-score | OSCR | Acc | H-score | OSCR | Acc | H-score | OSCR |
| NPN Sheng et al. (2024) | 71.08 | 24.46 | 61.45 | 54.58 | 41.21 | 38.79 | 55.92 | 42.95 | 35.03 | 51.97 | 11.28 | 34.23 | 58.39 | 29.98 | 42.38 |
| BadLabel Zhang et al. (2024) | 53.00 | 38.77 | 43.34 | 33.94 | 27.79 | 30.06 | 37.64 | 44.59 | 33.47 | 27.61 | 20.90 | 26.18 | 38.05 | 33.01 | 33.26 |
| ODGNet Bose et al. (2023) | 53.67 | 48.23 | 41.31 | 39.36 | 27.19 | 28.08 | 51.17 | 24.92 | 34.91 | 39.64 | 5.51 | 25.32 | 45.96 | 26.46 | 32.41 |
| MLDG Shu et al. (2019) | 68.17 | 23.34 | 55.68 | 56.47 | 40.89 | 40.97 | 56.31 | 41.94 | 38.02 | 47.81 | 11.35 | 31.60 | 57.19 | 29.38 | 41.57 |
| MEDIC-cls Wang et al. (2023) | 19.86 | 14.41 | 11.22 | 19.75 | 15.93 | 8.03 | 9.83 | 8.28 | 2.88 | 16.31 | 5.78 | 7.36 | 16.44 | 11.10 | 7.37 |
| MEDIC-bcls Wang et al. (2023) | 19.86 | 21.31 | 11.22 | 19.75 | 14.66 | 6.73 | 9.83 | 8.82 | 2.58 | 16.31 | 13.40 | 7.39 | 16.44 | 14.55 | 6.98 |
| EBiL-HaDS-cls Peng et al. (2024b) | 67.39 | 36.21 | 53.76 | 44.14 | 35.30 | 29.61 | 60.86 | 46.15 | 38.81 | 51.89 | 25.34 | 31.35 | 56.07 | 35.75 | 38.38 |
| EBiL-HaDS-bcls Peng et al. (2024b) | 67.39 | 44.88 | 50.01 | 44.14 | 36.20 | 37.31 | 60.86 | 48.40 | 41.70 | 51.89 | 23.25 | 36.46 | 56.07 | 38.18 | 41.37 |
| HyProMeta Peng et al. (2024a) | 73.53 | 50.23 | 61.08 | 60.42 | 46.38 | 46.23 | **69.81** | **54.72** | **50.39** | **57.28** | 18.84 | **37.45** | 65.26 | **42.54** | 48.79 |
| Ours | **84.80** | **56.59** | **71.81** | **62.44** | **47.95** | **47.72** | 67.36 | 51.08 | 47.78 | 55.55 | 12.27 | 36.56 | **67.54** | 41.97 | **50.97** |

Table 15: Results (%) of DigitsDG on ConvNet Zhou et al. (2021), where asymmetric label noise with ratio 50% is selected.

| Method | 20% sym | | | 50% sym | | | 80% sym | | | 50% asym | | |
|---|---|---|---|---|---|---|---|---|---|---|---|---|
| | Acc | H-score | OSCR | Acc | H-score | OSCR | Acc | H-score | OSCR | Acc | H-score | OSCR |
| HyProMeta Peng et al. (2024a) | 56.61 | 37.75 | 28.86 | 47.90 | 18.56 | 30.64 | 34.72 | 20.82 | 24.14 | 36.77 | 15.19 | 24.71 |
| Ours | **58.47** | **37.86** | **30.25** | **50.10** | **33.99** | **33.50** | **49.55** | **22.89** | **32.40** | **40.73** | **37.49** | **28.50** |

Table 16: Experimental results on TerriaINC dataset from DomainBed.

label. We compare our method with two variants, *i.e.*, GMM and FINCH, where we directly apply GMM and FINCH on the recorded loss to achieve binary clustering. From the experimental results, we can observe that our approach generally outperforms those two variants. FINCH Sarfraz et al. (2019) shows comparable performance with our approach on the symmetric label noise ratio of 20%, while our approach outperforms FINCH by large margins on the other label noise settings, demonstrating that the combination of the FINCH and GMM classifier is more robust to severe label noise. We further deliver more analysis for the sensitivity of the proposed HyProMeta to the clean/noisy partition. On PACS with art painting as the target domain and 50% label noise, reducing clustering accuracy from 92.25% to 42.76% and leading to a smaller OSCR drop from 59.58% to 46.97%. While performance is affected, the method remains robust due to selective clean sample usage, evidential pseudo-labeling, and meta-learning regularization. Although DBSCAN and KMEANS are also applied to loss trajectories, they remain highly sensitive to density assumptions and centroid initialization, which often leads to unstable cluster boundaries when loss patterns vary across domains and categories. In contrast, FINCH produces data-driven hierarchical partitions that do not require predefined density thresholds or cluster numbers, allowing it to better adapt to the heterogeneous and noisy loss dynamics characteristic of OSDG-NL. The subsequent GMM refinement further models the aggregated loss statistics with a probabilistic mixture, yielding a smoother and more discriminative clean/noisy separation than the rigid partitions produced by DBSCAN or KMEANS. Consequently, our FINCH+GMM pipeline delivers a more reliable clean subset, which directly strengthens the downstream residual-flow meta-learning process and leads to superior overall performance. We further provide the t-SNE visualizations where blue points denote correctly identified noisy samples and red points denote misclassified ones. On PACS (*Photo* as the target domain, 50% symmetric noise), our method achieves 92.25% label-noise detection accuracy, compared with 72.46% for HyProMeta Peng et al. (2024a), demonstrating a substantially more reliable separation in Figure 3 left hand side. The ablation of the epochs required for UTS-ELC is proposed in Figure 3 right hand side.

| Method | 20% sym | | | 50% sym | | | 80% sym | | | 50% asym | | |
|---|---|---|---|---|---|---|---|---|---|---|---|---|
| | Acc | H-score | OSCR | Acc | H-score | OSCR | Acc | H-score | OSCR | Acc | H-score | OSCR |
| MEDIC-cls Wang et al. (2023) | 41.74 | 0.42 | 32.57 | 24.46 | 6.54 | 16.93 | 10.06 | 3.52 | 6.44 | 27.30 | 9.83 | 19.35 |
| MEDIC-bcls Wang et al. (2023) | 41.74 | 37.58 | 31.60 | 24.46 | 23.38 | 17.38 | 10.06 | 12.69 | 6.52 | 27.30 | 27.87 | 20.01 |
| HyProMeta Peng et al. (2024a) | 42.32 | 40.35 | 33.86 | 25.58 | 28.91 | 20.11 | 12.40 | 16.00 | 8.76 | 27.55 | 28.25 | 21.60 |
| Ours | 44.07 | 42.71 | 36.27 | 41.61 | 40.53 | 32.51 | 20.08 | 22.43 | 13.07 | 37.65 | 34.75 | 28.11 |

Table 17: Experimental results on OfficeHome dataset.

| Method | 20% sym | | | 50% sym | | | 80% sym | | | 50% asym | | |
|---|---|---|---|---|---|---|---|---|---|---|---|---|
| | Acc | H-score | OSCR | Acc | H-score | OSCR | Acc | H-score | OSCR | Acc | H-score | OSCR |
| MEDIC-cls Wang et al. (2023) | 89.51 | 60.33 | 71.33 | 51.75 | 0.00 | 30.48 | 21.68 | 10.44 | 9.46 | 58.04 | 21.15 | 31.70 |
| MEDIC-bcls Wang et al. (2023) | 89.51 | 67.44 | 72.23 | 51.75 | 31.04 | 24.74 | 21.68 | 15.85 | 9.78 | 58.04 | 14.16 | 31.82 |
| HyProMeta Peng et al. (2024a) | 90.81 | 54.35 | 56.34 | 74.83 | 67.92 | 63.33 | 23.78 | 24.39 | 16.22 | 72.73 | 34.20 | 36.48 |
| Ours | 95.80 | 65.56 | 81.53 | 80.42 | 65.94 | 66.99 | 29.37 | 30.31 | 21.11 | 76.92 | 44.79 | 47.77 |

Table 18: Experimental results on VLCS dataset.

| Method | 20% sym | 50% sym | 80% sym | 50% asym |
|---|---|---|---|---|
| GMM | 72.11 | 87.83 | 54.56 | 50.49 |
| FINCH | 90.39 | 85.29 | 50.02 | 38.71 |
| DBSCAN | 82.42 | 54.45 | 20.12 | 24.71 |
| KMEANS | 80.05 | 50.08 | 20.12 | 24.71 |
| Representation | 54.80 | 49.74 | 45.32 | 25.07 |
| Ours | 90.04 | 92.25 | 56.05 | 52.91 |

Table 19: Ablation experiments on PACS *art painting* using ResNet18 He et al. (2016) as backbone for the unsupervised clustering approach regarding the label-clean/noisy sets partition. The performance is evaluated by the accuracy computed over the partitioned sample set using a binary indicator of whether the uncleaned label matches the original label for each sample.

| Method | #Params | PACS (Photo) | | | DigitsDG (mnist) | | |
|---|---|---|---|---|---|---|---|
| | | Acc | H-score | OSCR | Acc | H-score | OSCR |
| DiT-S | 30.98M | 77.71 | 75.29 | 70.07 | 74.86 | 4.45 | 63.42 |
| DiT-B | 129.60M | 82.39 | 81.52 | 78.68 | 85.97 | 64.79 | 69.88 |
| DiT-L | 435.90M | 77.46 | 65.01 | 63.65 | 78.83 | 22.96 | 66.43 |

Table 20: Ablation regarding the scalability of DC-CRFM using different sizes of DiT. Experiments are conducted on PACS dataset (test domain: *Photo*) and DigitsDG dataset (test domain: *MNIST*).

# G   TRAINING OVERHEAD AND COMPUTATION COST OF DC-CRFM

The number of parameters of our method is $\sim 215.6M$ during training, where DC-CRFM takes $\sim 129.6M$ due to its encoder-decoder structure for the generation of residuals, and $\sim 86.0M$ during testing when we use ViT-Base Dosovitskiy et al. (2021) as backbone, since DC-CRFM only participates in training. The whole training procedure takes $\sim 5h$ on PACS when we use one A100 GPU and ViT-Base as backbone.

# H   FURTHER CLARIFICATION REGARDING THE GENERALIZABILITY TO OTHER DATASETS

We conduct further experiments on TerraInc dataset Beery et al. (2018) from DomainBed Peng et al. (2019) with open-set ratio (8:2). The results are reported in Tab. 16, where we find our approach still outperforms the current best approach, HyProMeta. Across all noise conditions, the proposed method (Ours) outperforms HyProMeta in all metrics. Notably, under $50\%$ symmetric noise, it achieves a significant H-score gain of $15.43\%$ and OSCR gain of $2.86\%$, indicating improved robustness in separating clean/noisy samples and generalizing to unseen categories. Even under high-noise settings ($80\%$ symmetric and $50\%$ asymmetric), our method maintains superior OSCR and H-score, validating its effectiveness in tackling the OSDG-NL task.

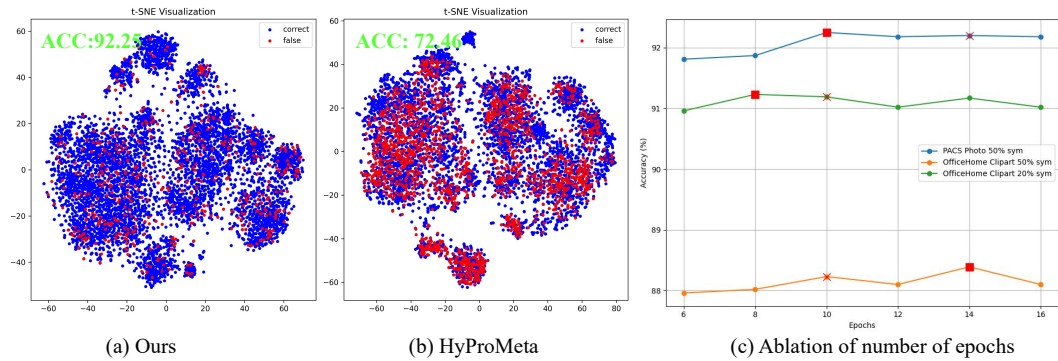

(a) Ours       (b) HyProMeta       (c) Ablation of number of epochs

Figure 3: (a) T-SNE visualizations of the clean/noisy partition performance of our approach, where the red dot denotes false separation and the blue dot denotes correct separation. (b) TSNE visualization of the clean/noisy partition performance of HyProMeta. (c) Ablation of the hyperparameter $N_e$ on PACS and OfficeHome dataset, where *Photo* and *Clipart* are chosen as target domains and label noise ratio is selected as $50\%$ symmetric.

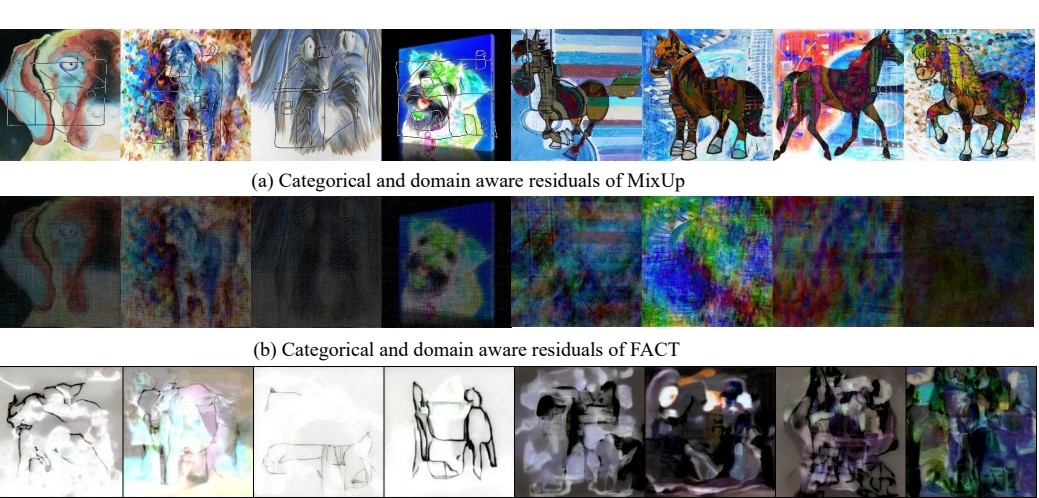

(a) Categorical and domain aware residuals of MixUp

(b) Categorical and domain aware residuals of FACT

(c) Categorical and domain aware residuals of our method

Figure 4: Visualization of domain and categorical residuals. The comparison is made between (a) categorical and domain-aware residuals of the MixStyle Zhou et al. (2020c) method, (b) categorical and domain-aware residuals of FACT Xu et al. (2021), and (c) categorical and domain-aware residuals of our proposed DC-CRFM. For each row, the first four figures follow the setting: (source domain: *Sketch*, target domain: *Art Painting*, source class: *Dog*, target class: *House*), while the rest four figures follow the setting: (source domain: *Cartoon*, target domain: *Art Painting*, source class: *Horse*, target class: *Guitar*).

# I   SCALING OF DIT

We provide the scalability evaluation in Tab. 20, where we find that DiT-B Peebles & Xie (2023) works the best compared to DIT-S/L across different datasets, and we also adopt DiT-B in our experiments. For PACS on the test domain *Photo*, DiT-B achieves the best results (Acc: $82.39\%$, H-score: $81.52\%$, OSCR: $78.68\%$), showing that scaling from DiT-S to DiT-B improves performance. However, further increasing the model size to DiT-L results in performance degradation, especially in H-score and OSCR.

For DigitsDG on the test domain *MNIST*, the gap is even more pronounced. DiT-B again performs the best (Acc: $85.97\%$, H-score: $64.79\%$, OSCR: $69.88\%$), whereas DiT-L suffers a sharp drop in H-score ($22.96\%$) despite having the highest parameter count. This indicates that DiT-B offers the best balance between model complexity and generalization for DC-CRFM.

| Method | | 20% sym | | | 50% sym | | | 80% sym | | | 50% asym | |
| | ACC | H-Score | OSCR | ACC | H-Score | OSCR | ACC | H-Score | OSCR | ACC | H-Score | OSCR |
|---|---|---|---|---|---|---|---|---|---|---|---|---|
| HyProMeta (mean) | 65.37 | 73.6 | 63.45 | 59.72 | 62.64 | 54.44 | 51.44 | 44.64 | 40.68 | 54.82 | 59.89 | 50.53 |
| HyProMeta (var) | ± 0.44 | ± 2.24 | ± 0.36 | ± 2.17 | ± 7.11 | ± 5.43 | ± 0.78 | ± 6.95 | ± 1.46 | ± 4.11 | ± 1.29 | ± 2.20 |
| Ours (mean) | 83.87 | 80.24 | 79.28 | 80.55 | 77.85 | 76.82 | 53.7 | 53.15 | 46.78 | 69.91 | 65.98 | 65.28 |
| Ours (var) | ± 1.43 | ± 0.68 | ± 3.09 | ± 0.84 | ± 2.98 | ± 1.57 | ± 2.47 | ± 3.48 | ± 1.45 | ± 0.98 | ± 1.33 | ± 1.71 |

Table 21: Statistical significance Results of our method and HyProMeta Peng et al. (2024a) on the PACS dataset when we select *Photo* as the target domain.

## J  VISUALIZATIONS OF LEARNED RESIDUALS

We further provide the visualizations of cross-domain and -category residuals in Figure 4. The categorical and domain residuals calculated based on the linear interpotation method proposed by MixStyle produces only linear transfer paths between pairs of samples, resulting in abrupt and visually incoherent transitions that can not capture more diverse and smooth domain or category shifts, while this limitation also exits for FACT Xu et al. (2021) as it achieves data augmentation by using linear interpolation in frequency domain, which does not explicitly model diverse and smooth transfer paths among diverse categories and domains. Note that we visualized the cross category and domain residuals, while during training only cross domain augmentation is used for MixStyle in our main experiments to ensure consistency with their original approach for domain generalization.

Because linear interpolation method directly depends on the finite set of available training samples, the types of residuals it can generate are fundamentally limited by the dataset scale, restricting the diversity and richness of cross-domain transformations. In contrast, our DC-CRFM learns structured residual distributions conditioned on domain and category labels, enabling smooth, soft, and semantically coherent transitions that better reflect true domain- and category-level variations. This benefit becomes especially important under label noise, where the clean/noisy separation cannot be perfectly accurate; in such cases, hard linear mixup method, *e.g.*, MixStyle Zhou et al. (2020c), often amplifies label corruption, while our flow-based residuals provide smooth and diverse image space transfers. By modeling continuous probability-flow trajectories rather than relying on linear interpolation, our method generates diverse and robust residuals that remain informative even when supervision is imperfect. Overall, DC-CRFM advances Mixstyle Zhou et al. (2020c) by offering smoother, more expressive, and distribution-level residual transformations that substantially improve domain generalization in noisy-label settings.

## K  PRELIMINARIES

In this section, we further provide an overview of the foundational components leveraged in our proposed method, including FINCH Sarfraz et al. (2019), Gaussian Mixture Models (GMMs), and vanilla Flow Matching.

### K.1  FINCH BASED UNSUPERVISED CLUSTERING METHOD

First Integer Neighbor Clustering (FINCH) Sarfraz et al. (2019) is a parameter-free clustering method which is built based on the following rules: each sample is linked to its first nearest neighbor, and clusters emerge as the connected components of this induced graph.

Given a set of embeddings $\{\mathbf{x}_i\}_{i=1}^{N}$, let $d(\mathbf{x}_i, \mathbf{x}_j)$ denote the cosine distance between the embeddings. FINCH Sarfraz et al. (2019) constructs a directed graph by assigning to each embedding $\mathbf{x}_i$ its first nearest neighbor according to Eq. 8.

$$\text{NN}(i) = \arg\min_{j \neq i} d(\mathbf{x}_i, \mathbf{x}_j). \tag{8}$$

A cluster assignment is achieved through grouping samples according to the transitive closure of this relation. Formally, two embeddings $\mathbf{x}_i$ and $\mathbf{x}_j$ belong to the same cluster $C_k$ if there exists a sequence according to Eq. 9.

$$\mathbf{x}_i \to \mathbf{x}_{a_1} \to \cdots \to \mathbf{x}_{a_m} \to \mathbf{x}_j, \tag{9}$$

such that each arrow represents a first-neighbor link according to Eq. 10.

$$\text{NN}(i) = a_1, \quad \text{NN}(a_1) = a_2, \quad \ldots, \quad \text{NN}(a_m) = j. \tag{10}$$

Clusters $\{C_k\}_{k=1}^K$ are therefore the connected components of the graph by Eq. 11,

$$G = (V, E), \qquad V = \{1, \ldots, N\}, \quad E = \{(i, \text{NN}(i))\}. \tag{11}$$

FINCH Sarfraz et al. (2019) applies this procedure hierarchically. Once the first-level clusters are obtained, each cluster $C_k$ is represented by its centroid:

$$\mu_k = \frac{1}{|C_k|} \sum_{\mathbf{x}_i \in C_k} \mathbf{x}_i, \tag{12}$$

and the algorithm repeats the nearest-neighbor linking step on the set of cluster centroids. This produces a sequence of increasingly coarse partitions $\left[\mathcal{C}^{(1)}, \mathcal{C}^{(2)}, \ldots, \mathcal{C}^{(L)}\right]$ until all samples merge into a single cluster.

Because FINCH Sarfraz et al. (2019) does not require the number of clusters nor density thresholds, and because its hierarchical structure naturally reveals coarse and fine partitions, it is well-suited for our clean/noisy separation based on evidential-loss trajectories.

## K.2    GAUSSIAN MIXTURE MODELS

Gaussian Mixture Models (GMMs) are probabilistic models that represent the feature distribution as a weighted sum of $K$ Gaussian components, as shown in Eq. 13

$$p(\mathbf{x}) = \sum_{k=1}^K \pi_k \, \mathcal{N}(\mathbf{x} \,|\, \mu_k, \Sigma_k), \tag{13}$$

where $\pi_k$ are weights for the mixture, and $\mathcal{N}(\cdot)$ denotes a Gaussian distribution with mean $\mu_k$ and covariance $\Sigma_k$. The parameters are usually learned using the Expectation-Maximization (EM) algorithm. GMMs provide soft probabilistic assignments, allowing us to refine clean/noisy separation by modeling uncertainty and distribution overlap in evidential-loss trajectories.

## K.3    FLOW MATCHING

Flow Matching (FM) is a generative method that learns a continuous-time velocity field which can transport samples from a source distribution to a target distribution. Instead of learning a score function or a diffusion procedure, FM directly estimates the vector field that describes how samples should move over time.

Given a pair of distributions $p_0(x)$ (source) and $p_1(x)$ (target), Flow Matching defines a family of intermediate distributions $p_t(x)$ generated by a time-dependent ordinary differential equation (ODE), according to Eq. 14.

$$\frac{d\mathbf{x}(t)}{dt} = \mathbf{v}_\gamma(\mathbf{x}(t), t), \qquad t \in [0, 1], \tag{14}$$

where $\mathbf{v}_\gamma$ is a learnable velocity field parameterized by $\gamma$. A solution trajectory $\mathbf{x}(t)$ of this ODE connects a source sample to a target sample, according to Eq. 15.

$$\mathbf{x}(0) \sim p_0, \qquad \mathbf{x}(1) \sim p_1. \tag{15}$$

To train the velocity field, FM constructs synthetic training trajectories using a straight-line path interpolation by Eq. 16.

$$\mathbf{x}_t = (1 - t)\,\mathbf{x}_0 + t\,\mathbf{x}_1, \tag{16}$$

where $\mathbf{x}_0 \sim p_0$ and $\mathbf{x}_1 \sim p_1$. The true (oracle) velocity associated with this path is as Eq. 17.

$$\mathbf{u}_t(\mathbf{x}_0, \mathbf{x}_1) = \mathbf{x}_1 - \mathbf{x}_0, \tag{17}$$

which is constant along the trajectory.

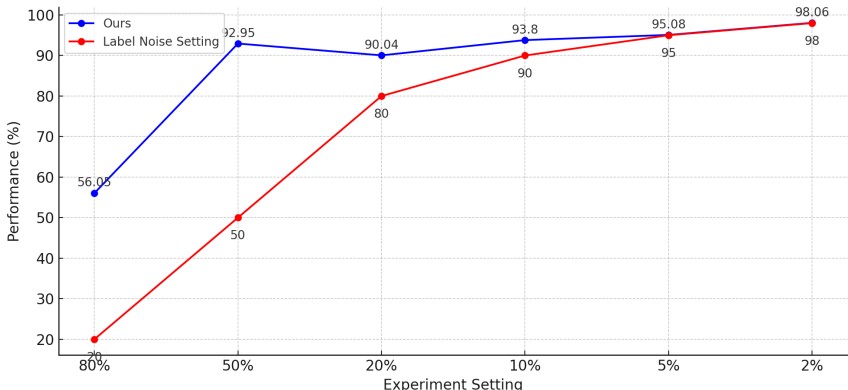

Figure 5: Ablation results for UTS-ELC for different label noise ratios on PACS dataset when we select *Photo* as the target domain.

The FM objective minimizes the squared error between the predicted velocity field and this oracle velocity as in Eq. 18.

$$\mathcal{L}_{\text{FM}}(\gamma) = \mathbb{E}_{\mathbf{x}_0 \sim p_0, \, \mathbf{x}_1 \sim p_1, \, t \sim \mathcal{U}(0,1)} \left[ \| \mathbf{v}_\gamma(\mathbf{x}_t, t) - (\mathbf{x}_1 - \mathbf{x}_0) \|^2 \right]. \tag{18}$$

Once trained, the velocity field defines a generative mapping. New samples can harvested by integrating the learned ODE according to Eq. 19.

$$\mathbf{x}(1) = \mathbf{x}(0) + \int_0^1 \mathbf{v}_\gamma(\mathbf{x}(t), t) \, dt. \tag{19}$$

## L    ANALYSIS OF ERROR PROPOGATION

In this section, we provide a detailed analysis of the error propagation behavior in our proposed framework. Understanding how misclassification between clean and noisy sets affects different components of the training pipeline is essential for explaining both the robustness and the limitations of UTS-ELC.

When a clean sample is mistakenly assigned to the noisy set, its impact on training is relatively mild. Such a sample is excluded from the meta-train pool, reducing its direct influence on the inner-loop optimization. However, it still participates in the meta-test stage, where its label information is utilized through both the original annotation and the evidential pseudo-label. As a result, the sample continues to contribute useful gradient signals during meta-test correction. Although this misplacement introduces some inconsistency, the meta-test supervision largely compensates for it, preventing substantial degradation. Consequently, this type of error leads to only limited error propagation throughout the training process.

A more detrimental situation arises when the opposite misclassification occurs—that is, when a noisy sample is incorrectly included in the clean set. In this scenario, the incorrect label is treated as reliable and is fed directly into the meta-train step. This is problematic because the meta-train stage lies at the core of the inner-loop optimization, meaning any erroneous gradient signals generated here will propagate through multiple updates. These corrupted gradients influence not only the immediate optimization but also subsequent meta-updates, amplifying their negative impact. This failure mode becomes particularly pronounced under extremely high noise rates, where the proportion of mislabeled samples in the clean set increases, causing unreliable supervision to dominate the learning process. This explains the noticeable performance degradation observed in such extreme noise conditions.

Despite these challenges, UTS-ELC remains consistently more robust than all baseline methods. Its dual-stage supervision, evidential modeling, and unified training scheme allow it to tolerate a considerable amount of noise before significant degradation occurs. Even under highly adverse

scenarios, the interplay between meta-test correction and evidential uncertainty estimation prevents catastrophic collapse, demonstrating the inherent resilience of the framework.

## M  ANALYSIS OF STATISTICS SIGNIFICANCE

In order to validate the statistic significance of our proposed approach, we further provide mean and standard error of five different runs of HyProMeta Peng et al. (2024a) and our proposed approach in Table 21. The results show that EReLiFM generally achieves the highest mean OSCR with smaller fluctuations of OSCR, even under strong label noise. In contrast, HyProMeta exhibits much larger variance, especially at higher noise ratios.

## N  ANALYSIS OF PSEUDO LABEL QUALITY

We further provide the analysis of the pseudo-label accuracy on PACS when *Photo* domain is used as the target domain. The pseudo-label accuracy and the corresponding final OSCR values are shown in Table.

From these results, we can see a clear observation: when the pseudo-label accuracy drops significantly, *e.g.*, under 80% symmetric noise or 50% asymmetric noise, the final OSCR also decreases. In contrast, when the pseudo-label accuracy stays reasonably high (around 77% or above, as in the 20% and 50% symmetric noise settings), its impact on OSCR is quite small. For example, these settings still achieve 78.68% and 77.52% OSCR, respectively.

Overall, this shows that the final performance is closely tied to the quality of pseudo-labels: once their accuracy falls extremly lower, the errors start to propagate during meta-testing and lead to noticeable performance drops. We acknowledge it as an open challenge for OSDG-NL and the above analysis is added into our revised paper.

| Method | 20% sym | 50% sym | 80% sym | 50% asym |
|---|---|---|---|---|
| $y_{\text{pseudo}}$ ACC | 90.34 | 77.23 | 32.90 | 75.48 |
| Final OSCR | 78.68 | 77.52 | 47.16 | 64.07 |

Table 22: Accuracy of the pseudo label prediction and the corresponding OSCR. Performances are reported on PACS dataset when *Photo* is selected as target domain.

## O  FURTHER JUSTIFICATION OF THE META LEARNING DESIGN

Our work mainly targets OSDG scenarios with significant and realistic label noise, where separating clean and noisy samples is both necessary and effective. Nevertheless, our UTS-ELC is designed to remain stable even when the noise level approaches 0%. As shown in Figure 5 of the appendix, the UTS-ELC drives the clean–noisy separation to naturally match the underlying noise ratio: when the noise ratio becomes small, the predicted noisy set also shrinks accordingly. On a fully clean dataset, the evidential-loss trajectories converge rapidly to low and stable values, causing almost all samples to be assigned to the clean set while the noisy set vanishes, as you said.

This behavior also ensures that meta-learning remains well-posed in the clean-data regime. The meta-test step is used only for label-correction under noisy supervision; thus, when the noisy set tends to zero, the optimization reduces to using the meta-train step alone. In this case, the model effectively collapses to standard supervised open-set domain generalization approach, which is sufficient for providing fully correct supervision.

However, the main focus of this work is on achieving reliable meta-learning under significant label noise and on mitigating the effect of severe label corruption during training as much as possible. We acknowledge that jointly achieve OSDG and OSDG-NL is important, and we are willing to consider it as future work.

