# OpenReview forum: "EReLiFM: Evidential Reliability-Aware Residual Flow Meta-Learning for Open-Set Domain Generalization under Noisy Labels"
_ICLR.cc/2026/Conference — Submitted to ICLR 2026_

### Official Review · Reviewer_Xk5Q · 2025-10-23

**Soundness:** 3
**Presentation:** 2
**Contribution:** 2
**Rating:** 4
**Confidence:** 4

**Summary:**

This paper addresses Open-Set Domain Generalization under Noisy Labels (OSDG-NL), where models must recognize unseen categories in new domains while training data contains label noise. The authors propose EReLiFM, which consists of three main components: (1) Unsupervised Two-Stage Evidential Loss Clustering (UTS-ELC) to separate clean from noisy samples using evidential loss trajectories, (2) Domain and Category Conditioned Residual Flow Matching (DC-CRFM) to generate diverse transfer paths across domains and categories, and (3) a meta-learning framework that trains on clean/augmented data and cautiously incorporates noisy samples with pseudo-labels. Experiments on PACS, DigitsDG, and TerraINC show improvements over the baseline HyProMeta.

**Strengths:**

1. **Important problem**: OSDG-NL is a relevant and challenging problem combining multiple realistic constraints.

2. **Comprehensive experiments**: The paper evaluates on multiple datasets, noise types (symmetric/asymmetric), and noise ratios (20%/50%/80%).

3. **Consistent improvements**: Results show improvements across most settings, though statistical significance is unclear.

4. **Thorough ablations**: Table 7 provides ablations of major components, though clarity could be improved.

**Weaknesses:**

1. **Weak theoretical justification**: Why should evidential loss trajectories be better than standard loss for clean/noisy separation? The paper provides intuition but no theoretical analysis or proof.

2. **Computational overhead**: Adding DiT (129.6M params) during training significantly increases computational cost. The paper doesn't analyze whether simpler augmentation strategies could achieve similar results.

3. **Failure mode analysis**: What happens when UTS-ELC fails (as suggested by 52-56% accuracy in Table 17)? How does error propagate through the pipeline?

4. **Limited domain diversity**: PACS has only 4 domains with similar image statistics. More diverse domains (e.g., natural images → medical images) would better validate the approach.

5. **Hyperparameter sensitivity**: No analysis of sensitivity to key hyperparameters (N_e, number of GMM components, meta-learning rates, etc.).

6. **Comparison fairness**: Some baselines (TCL, NPN, BadLabel) are not designed for OSDG, making comparisons less meaningful. More fair comparisons would be against other meta-learning or domain generalization methods adapted for noisy labels.

**Questions:**

1. **UTS-ELC failure modes**: Given the low separation accuracy (~55%) under high noise, how does the method remain robust? Can you provide analysis of what happens when clean samples are misclassified as noisy?

2. **Flow matching vs. MixUp**: Can you provide theoretical or empirical evidence that learning residuals via flow matching provides fundamentally different augmentations than MixUp? Visualizations of generated samples would help.

3. **Ablation details**: Please clarify the exact difference between "w/o UTS-ELC in RFM" and "w/ UTS-LC in RFM" in Table 7.

4. **Cross-dataset generalization**: How does a model trained on PACS with DC-CRFM perform when directly tested on DigitsDG without retraining?

5. **Hyperparameter selection**: How were hyperparameters (especially N_e=10) chosen? Is this consistent across datasets and noise levels?

6. **Statistical significance**: Can you provide error bars or significance tests for the main results?

7. **Pseudo-label quality**: What is the accuracy of pseudo-labels y_pseudo in the meta-test stage? How does this correlate with final performance?

---

> ### Author Response · Authors · 2025-11-24
> **Author Response - Part I**
>
> > Weak theoretical justification: Why should evidential loss trajectories be better than standard loss for clean/noisy separation? The paper provides intuition but no theoretical analysis or proof.
>
> Thank you very much for your comment.
>
> **Evidential loss differs from standard cross-entropy by explicitly modeling both predictive evidence and the associated uncertainty. This provides a richer and more structured training-dynamics signal that better supports clean/noisy separation.** Under the evidential formulation, correctly labeled samples accumulate high evidence and low uncertainty, causing their loss trajectories to decrease rapidly and stabilize. Noisy samples, however, produce inconsistent or conflicting evidence across epochs, resulting in higher uncertainty and more volatile trajectories. This separation emerges because evidential learning penalizes not only misclassification, but also unwarranted confidence, making mislabeled samples incur larger and more persistent penalties.
>
> Standard cross-entropy does not include this uncertainty term and therefore cannot reliably distinguish samples whose losses overlap or fluctuate due to early-training randomness. **From an optimization perspective, evidential loss corresponds to a Dirichlet-based belief-update process: clean samples move toward high-evidence regions, while noisy samples remain in low-evidence regimes due to contradictory supervision. This naturally produces a geometric margin between the two groups in trajectory space, which clustering methods can exploit more effectively.**
>
> This theoretical motivation, grounded in the evidential deep learning framework, along with our empirical results, provides justification for why evidential trajectories are more suitable for clean/noisy separation. We have added this theoretical discussion to the revised version of the paper.
>
> The abovementioned theoretical analysis is added into our revised paper in Sec. 3.2.
>
> >Computational overhead: Adding DiT (129.6M params) during training significantly increases computational cost. The paper doesn't analyze whether simpler augmentation strategies could achieve similar results.
>
> Thank you very much for your comment! **In Table 7 of our main paper, we directly compare our DC-CRFM with MixUp method, where we replace DC-CRFM with MixUp in our meta learning pipeline, and the results show clear performance degradation when MixUp is used**. Specifically, our method improves OSCR by 2.42% on DigitsDG with MNIST as the target domain and by 17.58% when SYN is used as the target domain.
> **In Table 1, we  have further compared against ODGNet (GAN-based augmentation) and MixStyle (a MixUp-style domain generalization method), and our approach achieves substantial gains over both in our original paper**. These results collectively demonstrate that residual flow matching provides more effective and robust augmentation for the OSDG-NL setting, whereas simpler augmentation strategies fail to capture meaningful cross-domain variations. Finally, we clarify that DiT is used only as an auxiliary component during training and does not introduce any computational overhead at inference time.
>
> > Failure mode analysis: What happens when UTS-ELC fails (as suggested by 52-56% accuracy in Table 17)? How does error propagate through the pipeline?
> Thank you for your comment.
>
>
> Thank you for your comment. We agree that UTS-ELC is not perfect, and Table 17 shows that, under the most challenging settings, the clean/noisy separation accuracy can drop to around 52–56%. We provide the error propogation analysis as follows.
>
> **When a clean sample is mistakenly assigned to the noisy set, it is removed from the meta-train pool but remains in the meta-test stage, where it is supervised by both the original label and the evidential pseudo-label. In this case, the sample is not discarded, and the meta-test correction mitigates most of the negative impact, leading to limited error propagation.**
>
> **In contrast, when a noisy sample is incorrectly assigned to the clean set, the resulting incorrect supervision enters the meta-train step directly. This is the primary failure mode, as erroneous gradients generated at meta-train will propagate through the inner-loop updates and ultimately degrade performance. This explains the performance drop observed under extremely high noise, where unreliable labels dominate the clean set.**
>
> Although UTS-ELC is affected in these extreme cases, it remains consistently better than all baseline methods, as shown by OSCR/Acc improvements in Tables 1–6 and 16–17. This demonstrates that the evidential trajectories and meta-learning framework retain robustness even when separation is imperfect. **We have added Sec. L in the appendix of our revised paper.**

---

> ### Author Response · Authors · 2025-11-24
> **Author Response - Part II**
>
> >Limited domain diversity: PACS has only 4 domains with similar image statistics. More diverse domains (e.g., natural images → medical images) would better validate the approach.
>
> Thank you for the valuable suggestion. **Our experiments follow the standard domain generalization setting, where source and target domains must share the same label space. This prevents us from evaluating natural-image → medical-image transfer, as these datasets contain entirely different semantic categories (e.g., cat/dog vs. heart/liver), making the DG protocol inapplicable.** To strengthen our empirical validation, we have expanded our experiments to two additional benchmarks, which are **OfficeHome and VLCS**. These two datastes are widely adopted in the DG community and offer greater domain diversity than PACS. **As shown in Table R5-A and Table R5-B**, our approach achieves consistent state-of-the-art performance across these datasets under the OSDG-NL setting. These results demonstrate that our method generalizes well beyond the leveraged datasets in our previous paper and remains effective across more diverse domain distributions.
>
>
> **Table R5-A: Experimental results on OfficeHome dataset**
>
> | **Method** | **20% sym Acc** | **20% sym Hscore** | **20% sym OSCR** | **50% sym Acc** | **50% sym Hscore** | **50% sym OSCR** | **80% sym Acc** | **80% sym Hscore** | **80% sym OSCR** | **50% asym Acc** | **50% asym Hscore** | **50% asym OSCR** |
> |------------|------------------|----------------------|---------------------------|------------------|----------------------|---------------------------|------------------|----------------------|---------------------------|---------------------|------------------------|-----------------------------|
> | MEDIC-cls | 41.74 | 0.42 | 32.57 | 24.46 | 6.54 | 16.93 | 10.06 | 3.52 | 6.44 | 27.30 | 9.83 | 19.35 |
> | MEDIC-bcls | 41.74 | 37.58 | 31.60 | 24.46 | 23.38 | 17.38 | 10.06 | 12.69 | 6.52 | 27.30 | 27.87 | 20.01 |
> | HyProMeta | 42.32 | 40.35 | 33.86 | 25.58 | 28.91 | 20.11 | 12.40 | 16.00 | 8.76 | 27.55 | 28.25 | 21.60 |
> | **Ours** | **44.07** | **42.71** | **36.27** | **41.61** | **40.53** | **32.51** | **20.08** | **22.43** | **13.07** | **37.65** | **34.75** | **28.11** |
>
>
>
>
> **Table R5-B: Experimental results on VLCS dataset**
>
> | **Method** | **20% sym Acc** | **20% sym H-score** | **20% sym OSCR** | **50% sym Acc** | **50% sym H-score** | **50% sym OSCR** | **80% sym Acc** | **80% sym H-score** | **80% sym OSCR** | **50% asym Acc** | **50% asym H-score** | **50% asym OSCR** |
> |------------|------------------|----------------------|---------------------------|------------------|----------------------|---------------------------|------------------|----------------------|---------------------------|---------------------|------------------------|-----------------------------|
> | MEDIC-cls | 89.51 | 60.33 | 71.33 | 51.75 | 0.00 | 30.48 | 21.68 | 10.44 | 9.46 | 58.04 | 21.15 | 31.70 |
> | MEDIC-bcls | 89.51 | 67.44 | 72.23 | 51.75 | 31.04 | 24.74 | 21.68 | 15.85 | 9.78 | 58.04 | 14.16 | 31.82 |
> | HyProMeta | 90.81 | 54.35 | 56.34 | 74.83 | 67.92 | 63.33 | 23.78 | 24.39 | 16.22 | 72.73 | 34.20 | 36.48 |
> | **Ours** | **95.80** | **65.56** | **81.53** | **80.42** | **65.94** | **66.99** | **29.37** | **30.31** | **21.11** | **76.92** | **44.79** | **47.77** |

---

> ### Author Response · Authors · 2025-11-24
> **Author Response -Part III**
>
> >Comparison fairness: Some baselines (TCL, NPN, BadLabel) are not designed for OSDG, making comparisons less meaningful. More fair comparisons would be against other meta-learning or domain generalization methods adapted for noisy labels.
>
> Thank you for the thoughtful comment. We agree that not all noisy-label methods are applicable to the OSDG setting, and that fair comparisons should involve approaches aligned with the domain-generalization and meta-learning nature of our task. For this reason, our main paper already evaluates against a comprehensive set of OSDG and meta-learning baselines adapted for noisy label setting: ARPL, ODGNet, MLDG, SWAD, MixStyle, MEDIC, and EBiL-HaDS represent established domain generalization methods; MLDG, MEDIC, EBiL-HaDS, and HyProMeta serve as meta-learning baselines; and HyProMeta is the only existing method specifically designed for OSDG-NL. As shown in Tables 1–6 of the main paper and Tables 8–18 in the appendix (with Tables 17 and 18 newly added), our method consistently outperforms these strong and relevant baselines. We have also added a clarification in the revised paper outlining the categories and applicability of the baselines to make the comparison fairness more explicit. The following text are added in our revision in Sec. 4.4.
>
> **For fair evaluation under the OSDG-NL setting, we compare against baselines that are compatible with domain generalization methods. ARPL, ODGNet, MLDG, SWAD, MixStyle, MEDIC, and EBiL-HaDS are established open-set domain generalization methods, while MLDG, MEDIC, EBiL-HaDS, and HyProMeta also serve as meta-learning approaches from open-set domain generalization field. HyProMeta is the only existing method specifically designed for OSDG-NL and thus provides the most directly comparable baseline. Methods designed for closed-set noisy-label learning (e.g., TCL, NPN, BadLabel) are also included, not as direct OSDG baselines, but to provide additional evaluation from a label-noise learning perspective. This ensures that our comparisons cover OSDG, noisy-label learning, and OSDG-NL dimensions.**
>
>
> >Flow matching vs. MixUp: Can you provide theoretical or empirical evidence that learning residuals via flow matching provides fundamentally different augmentations than MixUp? Visualizations of generated samples would help.
>
> Thank you for this helpful question. **In our revision, we explicitly address the difference between flow-matching residuals and MixUp both visually and quantitatively. First, as shown in Figure 4 (appendix), the residuals produced by MixUp correspond to hard linear interpolations between pairs of samples, which often lead to abrupt, ghosted, and visually incoherent transitions that do not reflect realistic domain or category shifts. In contrast, the residuals learned by our DC-CRFM via flow matching form smooth, structured, and semantically coherent transformations conditioned on domain and category labels, indicating that the model has learned a continuous residual distribution rather than simple convex combinations of existing samples.**
>
> Empirically, **Table 7 demonstrates that replacing DC-CRFM with MixUp causes clear performance drops on DigitsDG (e.g., OSCR decreases by 2.42% on mnist and 17.58% on syn as target domains), showing that MixUp-style augmentation is not sufficient for OSDG-NL.** Additional comparisons in Table 1 against **MixStyle (a MixUp-based DG method)** and **ODGNet** (GAN-based augmentation) further confirm that our flow-matching residuals lead to consistently higher OSCR and accuracy across domains. **Conceptually, residual flow matching learns a vector field that approximates continuous probability-flow trajectories between domain/category-conditioned distributions, whereas MixUp is restricted to discrete linear paths between finite training samples; this richer transport structure explains why DC-CRFM can better model cross-domain/category variation under label noise.** We have clarified these points and pointed to the new visualizations and results in the revised manuscript in Sec. J in appendix of our revision.

---

> ### Author Response · Authors · 2025-11-24
> **Author Response - Part IV**
>
> >Ablation details: Please clarify the exact difference between "w/o UTS-ELC in RFM" and "w/ UTS-LC in RFM" in Table 7.
>
> Thank you very much for pointing this out. **We have revised the text to clarify the exact meaning of “w/o UTS-ELC in RFM’’ and “w/ UTS-LC in RFM’’ in Table 7. In the w/o UTS-ELC in RFM setting, residual flow matching is trained on the entire dataset without performing any clean/noisy separation; that is, all samples (clean and noisy) are treated uniformly when learning residuals, while the rest setting is kept as the same with our final method. In the w/ UTS-LC in residual flow matching setting, we still perform clean/noisy clustering using standard loss trajectories (UTS-LC), but the evidential learning loss is removed,where the clustering relies only on plain cross-entropy trajectories, without uncertainty modeling.**
> This ablation isolates the impact of evidential loss on clean/noisy partition quality and shows how our method behaves when the clustering signal becomes less reliable. The corresponding description is added into Sec. 4.8. Please refer to our revised paper.
>
>
> >Cross-dataset generalization: How does a model trained on PACS with DC-CRFM perform when directly tested on DigitsDG without retraining?
>
> Thank you for your comment. **DC-CRFM is a domain- and category-conditioned residual flow model, which requires conditioning on specific domain identifiers and class labels that share the same semantic space. The DC-CRFM trained on PACS learns residuals tied to PACS domains (e.g., photo, cartoon, sketch) and PACS categories (e.g., cat, dog, horse).** In contrast, DigitsDG consists of entirely different domains (e.g., synthetic digits vs. real digits) and different semantic labels (digits 0–9). Since the two datasets do not share either the domain space or the label space, the residual flows learned on PACS cannot be meaningfully applied to DigitsDG without retraining. Therefore, direct cross DG dataset evaluation is not applicable for our DC-CRFM module.
>
> >Hyperparameter selection: How were hyperparameters (especially N_e=10) chosen? Is this consistent across datasets and noise levels?
> >Hyperparameter sensitivity: No analysis of sensitivity to key hyperparameters (N_e, number of GMM components, meta-learning rates, etc.).
>
>
> Thank you very much for your comment.
>
> **We set the number of GMM components to 2 because the goal is to separate samples into clean and noisy label groups. Other choices can not align with our clean/noisy separation paradigm.** We clarify that in all of our experiments we set the number of component as 2.
>
> **For the epoch window \(N_e\), we conducted an ablation study (please refer to Figure 3 c in our appendix of our revised figure, where we use red rectangular to mark the max position and red cross to mark the second max position).** We observe that the performance improves as \(N_e\) increases but becomes saturated around \(N_e = 10\). Beyond this point, storing longer loss trajectories does not yield additional gains and only increases computation and memory usage. Therefore, setting \(N_e > 10\) is not meaningful for our method. We further examined the consistency of this choice across datasets. Experiments on PACS and OfficeHome under 50% symmetric label noise show that \(N_e = 10\) achieves the best performance when Photo is the target domain in PACS and yields the second-best results in most OfficeHome settings when Clipart is chosen as target domain. Given these performances and to ensure fair comparisons, we adopt a unified value of \(N_e = 10\) for all experiments. Meta learning rate is selected according to the performance on the validation set for source domains.

---

> ### Author Response · Authors · 2025-11-24
> **Author Response - Part V**
>
> > Statistical significance: Can you provide error bars or significance tests for the main results?
>
> Thank you very much for your comment. Thank you for the suggestion. **We have added statistical significance results in Table 21 of the appendix (corresponding to Table R5-C), reporting the mean and standard deviation across 5 runs.** The results show that EReLiFM generally achieves the highest mean OSCR with smaller fluctuations of OSCR, even under strong label noise. In contrast, HyProMeta exhibits much larger variance, especially at higher noise ratios.
> This demonstrates that the improvements of EReLiFM are not due to randomness but reflect stable and reproducible behavior. We have also added correponding description of this table in Sec. M. Please refer to our revised paper.
>
>
> **Table R5-C**
>
> | Method | 20% sym ACC | 20% sym H-Score | 20% sym OSCR | 50% sym ACC | 50% sym H-Score | 50% sym OSCR | 80% sym ACC | 80% sym H-Score | 80% sym OSCR | 50% asym ACC | 50% asym H-Score | 50% asym OSCR |
> |--------|-------------|------------------|---------------|--------------|------------------|---------------|--------------|------------------|---------------|----------------|-------------------|----------------|
> | **HyProMeta (mean)** | 65.37 | 73.60 | 63.45 | 59.72 | 62.64 | 54.44 | 51.44 | 44.64 | 40.68 | 54.82 | 59.89 | 50.53 |
> | **HyProMeta (var)** | ±0.44 | ±2.24 | ±0.36 | ±2.17 | ±7.11 | ±5.43 | ±0.78 | ±6.95 | ±1.46 | ±4.11 | ±1.29 | ±2.20 |
> | **Ours (mean)** | 83.87 | 80.24 | 79.28 | 80.55 | 77.85 | 76.82 | 53.70 | 53.15 | 46.78 | 69.91 | 65.98 | 65.28 |
> | **Ours (var)** | ±1.43 | ±0.68 | ±3.09 | ±0.84 | ±2.98 | ±1.57 | ±2.47 | ±3.48 | ±1.45 | ±0.98 | ±1.33 | ±1.71 |
>
>
> >Pseudo-label quality: What is the accuracy of pseudo-labels y_pseudo in the meta-test stage? How does this correlate with final performance?
>
> Thank you very much for your comments! **To address your question, we report the pseudo-label accuracy on PACS when photo domain is used as the target domain. The pseudo-label accuracy and the corresponding final OSCR values are shown in Table R5-D.**
>
> From these results, we can see a clear observation: when the pseudo-label accuracy drops significantly, e.g., under 80% symmetric noise or 50% asymmetric noise, the final OSCR also decreases. In contrast, when the pseudo-label accuracy stays reasonably high (around 77% or above, as in the 20% and 50% symmetric noise settings), its impact on OSCR is quite small. For example, these settings still achieve 78.68% and 77.52% OSCR, respectively.
>
> Overall, this shows that the final performance is closely tied to the quality of pseudo-labels: once their accuracy falls extremly lower, the errors start to propagate during meta-testing and lead to noticeable performance drops. We acknowledge it as an open challenge for OSDG-NL and the above analysis is added into our revised paper. Thank you for pointing this out. The corresponding description is added into Sec. N in the appendix of our revised paper, please refer to our revision.
>
> **Table R5-D**
> | Method       | 20 sym | 50 sym | 80 sym | 50 asym |
> |--------------|--------|--------|--------|---------|
> |y_pseudo ACC | 90.34  | 77.23  | 32.9   | 75.48   |
> | Final OSCR   | 78.68  | 77.52  | 47.16  | 64.07   |

---

### Official Review · Reviewer_FcXz · 2025-11-01

**Soundness:** 2
**Presentation:** 3
**Contribution:** 2
**Rating:** 4
**Confidence:** 4

**Summary:**

This paper studies the problem of open-set domain generalization under noisy labels. The proposed method consists of two modules: denoising (dataset separation) and training. In the denoising stage, the authors use Evidential Learning to estimate the uncertainty of samples, then apply a clustering algorithm to divide them into a clean set (low uncertainty clusters) and a noisy set (high uncertainty clusters). Training is conducted within a meta-learning framework, where the meta-train data come from the original and flow-model-augmented clean set, while the meta-test data are derived from the noisy set with pseudo-labels. Experimental results demonstrate the effectiveness of the proposed approach.

**Strengths:**

1. The paper is clear and easy to understand.
2. The application of Evidential Learning, though rarely seen in open-set recognition，seems reasonable.
3. Experimental results demonstrate the effectiveness of the proposed method.

**Weaknesses:**

1. The choice to use the clean set for meta-training and the noisy set for meta-testing appears odd. Performance gains can often be achieved through various tuned meta-learning strategies—whether traditional sampling from the same distribution or MEDIC’s domain-class sampling. Therefore, the authors should provide stronger justification that their approach offers unique advantages. Moreover, the meta-learning experiments should include comparisons across different meta-learning strategies, rather than only ablation studies (e.g., removing the pseudo-labeling module), since it is somewhat obvious that pseudo-labeling would help.
2. The proposed dataset separation strategy always produces a clean set and a noisy set. However, on a completely clean dataset, this could have adverse effects: if the noisy set is small, there would be insufficient meta-test data; if it is large, many correct samples could be incorrectly assigned pseudo-labels.
3. The paper repeatedly emphasizes the advantage of the flow model over Mixup interpolation, yet the evidence is limited to accuracy comparisons. Given the emphasis placed on this claim, more qualitative or intuitive analyses are needed to substantiate it. Furthermore, Mixup is not the only data augmentation approach worth comparing.
4. The studies cited in lines 104–105 are flow matching, and some studies referenced in lines 386–388 are close set domain generalization. They are not open set domain generalization.

**Questions:**

1. Please see weaknesses.
2. What is the motivation for studying open set domain generalization and noisy labels simultaneously? Do the authors assume that either open set domain generalization or domain generalization under noisy labels has already been well-studied?
3. How is y_a chosen in the algorithm?

---

> ### Author Response · Authors · 2025-11-24
> **Author Response - Part I**
>
> >The choice to use the clean set for meta-training and the noisy set for meta-testing appears odd. Performance gains can often be achieved through various tuned meta-learning strategies—whether traditional sampling from the same distribution or MEDIC’s domain-class sampling. Therefore, the authors should provide stronger justification that their approach offers unique advantages. Moreover, the meta-learning experiments should include comparisons across different meta-learning strategies, rather than only ablation studies (e.g., removing the pseudo-labeling module), since it is somewhat obvious that pseudo-labeling would help.
>
>
> Thank you for the insightful comment! We agree that meta-learning performance can be influenced by various sampling strategies, and that a stronger justification of our clean–noisy meta-train/meta-test design is necessary. **We would like to clarify that our original submission already includes comprehensive comparisons against four representative meta-learning baselines, i.e., EBiL-HaDS, MEDIC, MLDG, and HyProMeta, which together span a wide spectrum of meta-learning formulations, including instance-level consistency, domain–class sampling, distribution alignment, and prototype-guided clean-sample selection.** These baselines therefore already expand a broad range of alternative meta-learning strategies beyond pseudo-labeling.
>
> **Regarding the question for the choice of using clean set in meta training and the noisy set in the meta test, the goal of the meta train is to model a generalizable update rule of the training, which builds up the foundation of the parameter optimization of the network. Thereby in this stage we need clean and reliable supervision so that the meta-learner can learn stable, unbiased gradient directions.  While the goal of meta-test is to evaluate and refine the learned update rule under real conditions, e.g., label noise perturbation. Corrections (i.e., evidential pseudo-labeling) are applied here to refine predictions based on the meta-trained update dynamics.**
>
> This design choice is fundamentally different from domain-based sampling schemes (e.g., MEDIC, EBiL-HaDS) or prototype-based relabelling (e.g., HyProMeta), which assume that inner-loop supervision is reliable. Our updated Sec. 3.3 now highlights this rationale more explicitly, emphasizing that the goal of meta-learning here is not merely cross-domain adaptation but learning to generalize from reliable supervision (clean set) toward unreliable and corrected supervision (noisy set).
>
> The corresponding experimental results can be found in the main tables of our paper and our experiment analysis section. Our method consistently performs best across multiple datasets and noise levels. This demonstrates that the improvement is not due to sampling tricks or pseudo-labeling alone, but arises also from the principled separation between meta trian and meta test, together with the categorical and domain aware flow matching approach.
>
>
> >The proposed dataset separation strategy always produces a clean set and a noisy set. However, on a completely clean dataset, this could have adverse effects: if the noisy set is small, there would be insufficient meta-test data; if it is large, many correct samples could be incorrectly assigned pseudo-labels.
>
>
> Thank you for raising this important point.
> Our work mainly targets OSDG scenarios with significant and realistic label noise, where separating clean and noisy samples is both necessary and effective. Nevertheless, our UTS-ELC is designed to remain stable even when the noise level approaches 0%. **As shown in Figure 5 of the appendix, the UTS-ELC drives the clean–noisy separation to naturally match the underlying noise ratio: when the noise ratio becomes small, the predicted noisy set also shrinks accordingly.** On a fully clean dataset, the evidential-loss trajectories converge rapidly to low and stable values, causing almost all samples to be assigned to the clean set while the noisy set vanishes, as you said.
>
> **This behavior also ensures that meta-learning remains well-posed in the clean-data regime. The meta-test step is used only for label-correction under noisy supervision; thus, when the noisy set tends to zero, the optimization reduces to using the meta-train step alone. In this case, the model effectively collapses to standard supervised open-set domain generalization approach, which is sufficient for providing fully correct supervision.**
>
> However, the main focus of this work is on achieving reliable meta-learning under significant label noise and on mitigating the effect of severe label corruption during training as much as possible. We acknowledge that jointly achieve OSDG and OSDG-NL is important, and we are willing to consider it as future work. Thank you very much for your comment! The corresponding description is added into Sec. O in the appendix of our revised paper.

---

> ### Author Response · Authors · 2025-11-24
> **Author Response - Part II**
>
> >The paper repeatedly emphasizes the advantage of the flow model over Mixup interpolation, yet the evidence is limited to accuracy comparisons. Given the emphasis placed on this claim, more qualitative or intuitive analyses are needed to substantiate it. Furthermore, Mixup is not the only data augmentation approach worth comparing.
>
>
> Thank you very much for your comment! **We have added visualizations of the learned categorical and domain residuals in Figure 4(appendix). These results show that MixUp (adopted according to MixStyle method) generates only linear interpolations between pairs of samples, often producing abrupt and visually incoherent transitions that fail to capture real domain or category shifts.** Because MixUp is limited by the finite set of training samples, it cannot provide the diversity needed for robust cross-domain and categorcy transformations. **In contrast, our DC-CRFM learns structured smooth residual distributions conditioned on domain and category labels, enabling smooth and semantically coherent transitions that better reflect true cross-domain/category variation.**
>
> **We have also compared with ODGNet, which is an GAN based augmentation approach alreay existing in our Tables 1-6, where our approach consistently works well than both of ODGNet and MixStyle. Additional analysis is provided in Sec. J of the appendix.**
>
>
> >The studies cited in lines 104–105 are flow matching, and some studies referenced in lines 386–388 are close set domain generalization. They are not open set domain generalization.
>
> Thank you for your comments. In the related work section on open set domain generalization (OSDG), our intention was to clarify that OSDG combines two challenges: domain generalization and the open set setting. Therefore, we first included works on domain generalization, as this forms a foundational component of the OSDG task.
>
> Regarding the flow-matching citations, we included these studies because our method incorporates the concept of flow matching. To help readers understand the motivation and technical background of our approach, we first introduced and summarized relevant flow-matching works.
>
>
> > What is the motivation for studying open set domain generalization and noisy labels simultaneously? Do the authors assume that either open set domain generalization or domain generalization under noisy labels has already been well-studied?
>
> **The motivation for studying open-set domain generalization (OSDG) together with noisy labels is that real-world OSDG applications inevitably encounter both domain/category shifts and imperfect annotations.** Label noise corrupts the source-domain knowledge required to recognize known classes and reject unseen ones, making OSDG significantly more challenging. Although domain generalization and noisy-label learning have each been studied individually, their combination in the open-set setting has been largely overlooked. **As noted in the paper, noisy-label learning is well explored in standard classification, but remains largely unaddressed in OSDG.** HyProMeta is the first method specifically designed for OSDG with noisy labels, showing that the area is still emerging rather than solved. Therefore, our work does not assume this problem is well-addressed.
>
> >How is y_a chosen in the algorithm?
>
> Thank you for your question! **y_a is chosen as an additional category.** For example, on PACS dataset we have 6 known classes, in this case y_a is regarded as the 7-th class for the training.

---

### Official Review · Reviewer_WyRZ · 2025-11-01

**Soundness:** 2
**Presentation:** 2
**Contribution:** 2
**Rating:** 4
**Confidence:** 3

**Summary:**

The paper proposes EReLiFM, a three-stage pipeline for Open-Set Domain Generalization (OSDG) under noisy labels. The approach: (i) separates clean from noisy samples via UTS-ELC (Unsupervised Two-Stage Evidential Loss Clustering), (ii) enriches the training data using DC-CRFM (Domain- and Category-Conditioned Residual Flow Matching), and (iii) optimizes a meta-learning objective that decouples clean and noisy supervision. Experiments demonstrate strong performance on PACS, DigitsDG and TerraINC datasets.

**Strengths:**

Well-motivated pipeline. The experimental results demonstrate substantial improvements over existing baselines across multiple benchmarks.

**Weaknesses:**

1. The presentation is kind of poor. Instead of spending a whole page (i.e., page 3) presenting the motivation for all components, I think it is better to present a high-level framework, e.g.:

    - separates clean from noisy samples
    - enriches the training data
    - optimizes a meta-learning objective that decouples clean and noisy supervision

   Then, within each subsequent section for individual components, describe the specific motivation and highlight key differences from previous works. This structure may help readers grasp the overall strategy before exploring component-level details.

2. Meta-learning step relies heavily on plain text description. Provide explicit mathematical formulations showing the inner/outer loop objectives, gradient flows, and how clean vs. noisy samples are weighted.

3. Since the proposed method incorporates many existing techniques (Finch clustering, GMM, residual flow matching), adding a Background/Preliminaries subsection that concisely explains these foundational methods would make the paper accessible to readers less familiar with these techniques.

4. Experiments: Rather than relying solely on percentage improvements (which readers can see in the tables), consider including:

    - t-SNE or UMAP plots demonstrating UTS-ELC's ability to separate clean from noisy samples
    - Sample quality comparisons or interpolation visualizations showing DC-CRFM-generated data quality and diversity

5. Acronym introduction. UTS-ELC and DC-CRFM are used before being introduced (lines 64 and 66).

**Questions:**

Please see the weakness above.

---

> ### Author Response · Authors · 2025-11-24
> **Author Response**
>
> > The presentation is kind of poor. Instead of spending a whole page (i.e., page 3) presenting the motivation for all components, I think it is better to present a high-level framework, e.g.:...
>
> Thank you very much for your comment. **We have added Figure 1 in our revised paper to illustrate the whole high-level framework. We rewrote Sec. 3.2 to present a clearer, top-down explanation of our method following your suggestion, which indeed makes the presentation of our paper clearer.**
> The revised section begins with a high-level overview that outlines the three-stage pipeline, clean/noisy separation, residual flow–based data enrichment, and label noise aware meta-learning, before introducing the detailed motivations for each component.
> We first clarified how UTS-ELC leverages evidential-loss trajectories and domain/category awareness to improve clean–noisy separation. We then reorganized the DC-CRFM and meta-learning descriptions to highlight their respective novelty and their roles within the overall framework. Thank you for your great comments! Please refer to our revised paper.
>
> > Meta-learning step relies heavily on plain text description. Provide explicit mathematical formulations showing the inner/outer loop objectives, gradient flows, and how clean vs. noisy samples are weighted.
>
> Thank you for the insightful comment. We agree that the meta-learning step benefits from clearer mathematical grounding. **Our original paper already includes the explicit inner/outer loop objectives and gradient update rules in Algorithm 1 (Lines 13 and 17 in Alg 1) and we highlighted them in our revision and have added the weights for losses**. Please refer to our revised paper.
>
>
>
> > Since the proposed method incorporates many existing techniques (Finch clustering, GMM, residual flow matching), adding a Background/Preliminaries subsection that concisely explains these foundational methods would make the paper accessible to readers less familiar with these techniques.
>
> Thank you very much for your comments! **We have added additional Sec. K in our appendix for the preliminaries.** We replace residual flow matching into flow matching as residual flow matching is for the first time proposed by us in our main paper. Please refer to our revised paper.
>
>
>
> > Experiments: Rather than relying solely on percentage improvements (which readers can see in the tables), consider including:
>
> Thank you very much for your comment. **We have added t-SNE visualizations in Figure 3(a) and (b)**, where blue points denote correctly identified noisy samples and red points denote misclassified ones. On PACS (Photo as the target domain, 50% symmetric noise), **our method achieves 92.25% label-noise detection accuracy, compared with 72.46% for HyProMeta**, demonstrating a substantially more reliable separation. Corresponding description is added into Sec. F in the appendix of our revised paper.
>
> We have also included **the learned categorical and domain residuals in Figure 4 of the appendix**. These visualizations show that MixUp (from MixStyle) generates only linear, sample-pair interpolations, which often produce abrupt or incoherent transitions and cannot model meaningful domain or category shifts. In contrast, **our DC-CRFM learns smooth and structured residual distributions conditioned on domain and category labels, enabling semantically coherent cross-domain/category transformations**. This advantage is especially important under label noise: MixUp tends to amplify incorrect labels, whereas our flow-based residuals provide stable, uncertainty-aware augmentations. Additional analyses have been added to Sec. J of the appendix.
>
> >Acronym introduction. UTS-ELC and DC-CRFM are used before being introduced (lines 64 and 66).
>
> Thank you for pointing them our! We have modified them in our revised version.

---

### Official Review · Reviewer_HVGE · 2025-11-02

**Soundness:** 3
**Presentation:** 2
**Contribution:** 3
**Rating:** 4
**Confidence:** 2

**Summary:**

This paper attempts to tackle the problem of Open-Set Domain Generalization under Noisy Labels (OSDG-NL). The authors proposed an approach based on evidential loss and residual flow, named Evidential Reliability-Aware Residual Flow Meta-Learning (EReLiFM). EReLiFM includes two modules, named UTS-ELC and DC-CRFM, respectively. UTS-ELC promotes better clean/noise separation across domains. DC-CRFM could augment more data with structured residuals. Subsequently, these two modules are integrated within a meta-learning framework. Experiments demonstrates that EReLiFM could enhance the performance of noise diagnosis and data augment for OSDG-NL.

**Strengths:**

[+] The detail of the method and experiment is well described.

[+] The related work is detailed.

[+] The experiments conducted are extensive.

**Weaknesses:**

Major weakness:

[-] The authors propose UTS-ELC to better separate clean and noisy samples using evidential loss. Please clarify the mechanism and explain its advantage in achieving more reliable separation compared to state-of-the-art methods such as HyProMeta. In addition, please provide visualizations or other metrics to demonstrate this claimed advantage.

[-] The authors propose DC-CRFM to expand clean data with structured diversity. However, there is a lack of visual results illustrating the advantage of DC-CRFM in enhancing diversity compared to existing augmentation methods (e.g., MixStyle [1] and FACT [2]).

[-] The benchmarks used appear limited. Can the proposed approach consistently achieve better performance on commonly used domain generalization datasets such as OfficeHome and VLCS?

[1]. Domain Generalization with MixStyle. ICLR, 2020.

[2]. A Fourier-based Framework for Domain Generalization. CVPR, 2021.

Minor weakness:

[-] The manuscript requires further refinement in details to improve readability. For instance, "DirectFM" in Table 7 is not defined.

**Questions:**

Please refer to the weaknesses.

---

> ### Author Response · Authors · 2025-11-24
> **Author Response - Part I**
>
> >The authors propose UTS-ELC to better separate clean and noisy samples using evidential loss. Please clarify the mechanism and explain its advantage in achieving more reliable separation compared to state-of-the-art methods such as HyProMeta. In addition, please provide visualizations or other metrics to demonstrate this claimed advantage.
>
>
> Thank you very much for your suggestion! We have added **TSNE visualizations in Figure 3 (a) and (b) in our revised paper**, where the blue dot denotes correct label-noise detection and red dot denotes false label-noise detection.
> On PACS dataset, when we select Photo as target domain and symmetric label noise ratio as 50%, **our approach can achieve 92.25% accuracy while HyProMeta can achieve 72.46% accuracy for label noise detection**.
> UTS-ELC separates clean and noisy samples by clustering evidential-loss trajectories, which capture both prediction errors and uncertainty over the entire training process rather than relying on a single embedding snapshot.
> The ultilization of the training dynamics is more reliable under domain shift and label noise, enabling UTS-ELC to detect mislabeled samples more reliably than HyProMeta’s hyperbolic-prototype approach. Feature based separation is sensitive to outlier features, as feature-space embeddings can be highly distorted by noisy labels, causing clean and noisy samples to overlap and reducing clustering reliability.
> The two-stage structure (FINCH followed by GMM) refines the separation by modelling domain- and category-aware loss statistics, yielding a smoother and more discriminative partition. More comparisons with other unsupervised clustering methods based on loss trajectories are shown in Table 19, as attached in Table R2-A in this response.
>
>
> **Table R2-A**
> | Method            | 20% sym | 50% sym | 80% sym | 50% asym |
> |-------------------|---------|---------|---------|----------|
> | GMM               | 72.11   | 87.83   | 54.56   | 50.49    |
> | FINCH             | **90.39**   | 85.29   | 50.02   | 38.71    |
> | DBSCAN            | 82.42   | 54.45   | 20.12   | 24.71    |
> | KMEANS            | 80.05   | 50.08   | 20.12   | 24.71    |
> | Representation    | 54.80   | 49.74   | 45.32   | 25.07    |
> | **Ours**          | 90.04 | **92.25** | **56.05** | **52.91** |
>
>
>
> >The authors propose DC-CRFM to expand clean data with structured diversity. However, there is a lack of visual results illustrating the advantage of DC-CRFM in enhancing diversity compared to existing augmentation methods (e.g., MixStyle [1] and FACT [2]).
>
> Thank you for your comment!
>
> We have **visualized the categorical and domain residuals** in our revised paper (**Figure 4 in appendix**).
> The categorical and domain residuals calculated based on the linear interpotation method proposed by MixStyle produces only linear transfer paths between pairs of samples, resulting in abrupt and visually incoherent transitions that can not capture more diverse and smooth domain or category shifts, while this limitation also exits for FACT as it achieves data augmentation by using linear interpolation in frequency domain, which does not explicitly model diverse and smooth transfer paths among diverse categories and domains. Note that we visualized the cross category and domain residuals, while during training only cross domain augmentation is used for MixStyle in our main experiments to ensure consistency with their original approach for domain generalization.
>
> Because linear interpolation method directly depends on the finite set of available training samples, the types of residuals it can generate are fundamentally limited by the dataset scale, restricting the diversity and richness of cross-domain transformations.
> In contrast, our DC-CRFM learns structured residual distributions conditioned on domain and category labels, enabling smooth, soft, and semantically coherent transitions that better reflect true domain- and category-level variations. This benefit becomes especially important under label noise, where the clean/noisy separation cannot be perfectly accurate; in such cases, hard linear mixup method, e.g., MixStyle, often amplifies label corruption, while our flow-based residuals provide smooth and diverse image space transfers.
> By modeling continuous probability-flow trajectories rather than relying on linear interpolation, our method generates diverse and robust residuals that remain informative even when supervision is imperfect. Overall, DC-CRFM advances Mixstyle by offering smoother, more expressive, and distribution-level residual transformations that substantially improve domain generalization in noisy-label settings. The corresponding analysis has been added in Sec J of the appendix, please refer to our revised paper.

---

> ### Author Response · Authors · 2025-11-24
> **Author Response - Part II**
>
> >The benchmarks used appear limited. Can the proposed approach consistently achieve better performance on commonly used domain generalization datasets such as OfficeHome and VLCS?
>
> Thank you very much for your comment! **We have added new comparisons of our proposed approach and well performing baselines in Table 17 (corresponding to Table R2-B in rebuttal response) and Table 18 (corresponding to Table R2-C in rebuttal response) of our revised paper.**
>
> Across all noise settings on the OfficeHome dataset, our method achieves the best results, for example, under 50% symmetric noise, it outperforms OSCR from HyProMeta’s 20.11% to 32.51%, a gain of more than 12%. The advantage becomes even clearer under the challenging 80% symmetric noise setting, where our method reaches an OSCR of 13.07%, significantly higher than HyProMeta’s 8.76%. Under 50% asymmetric noise, our method also surpasses prior approaches, achieving an OSCR of 28.11% compared to HyProMeta’s 21.60%, demonstrating robust performance under both symmetric and asymmetric noise. Similar performance gains brought by our approach can also be observed in Table R2-C for the experiments on VLCS, which illustrates the great generalizability of our proposed approach across benchmarks.
>
> **Table R2-B: Experimental results on OfficeHome dataset**
>
> | **Method** | **20% sym Acc** | **20% sym Hscore** | **20% sym OSCR** | **50% sym Acc** | **50% sym Hscore** | **50% sym OSCR** | **80% sym Acc** | **80% sym Hscore** | **80% sym OSCR** | **50% asym Acc** | **50% asym Hscore** | **50% asym OSCR** |
> |------------|------------------|----------------------|---------------------------|------------------|----------------------|---------------------------|------------------|----------------------|---------------------------|---------------------|------------------------|-----------------------------|
> | MEDIC-cls | 41.74 | 0.42 | 32.57 | 24.46 | 6.54 | 16.93 | 10.06 | 3.52 | 6.44 | 27.30 | 9.83 | 19.35 |
> | MEDIC-bcls | 41.74 | 37.58 | 31.60 | 24.46 | 23.38 | 17.38 | 10.06 | 12.69 | 6.52 | 27.30 | 27.87 | 20.01 |
> | HyProMeta | 42.32 | 40.35 | 33.86 | 25.58 | 28.91 | 20.11 | 12.40 | 16.00 | 8.76 | 27.55 | 28.25 | 21.60 |
> | **Ours** | **44.07** | **42.71** | **36.27** | **41.61** | **40.53** | **32.51** | **20.08** | **22.43** | **13.07** | **37.65** | **34.75** | **28.11** |
>
>
>
>
> **Table R2-C: Experimental results on VLCS dataset**
>
> | **Method** | **20% sym Acc** | **20% sym H-score** | **20% sym OSCR** | **50% sym Acc** | **50% sym H-score** | **50% sym OSCR** | **80% sym Acc** | **80% sym H-score** | **80% sym OSCR** | **50% asym Acc** | **50% asym H-score** | **50% asym OSCR** |
> |------------|------------------|----------------------|---------------------------|------------------|----------------------|---------------------------|------------------|----------------------|---------------------------|---------------------|------------------------|-----------------------------|
> | MEDIC-cls | 89.51 | 60.33 | 71.33 | 51.75 | 0.00 | 30.48 | 21.68 | 10.44 | 9.46 | 58.04 | 21.15 | 31.70 |
> | MEDIC-bcls | 89.51 | 67.44 | 72.23 | 51.75 | 31.04 | 24.74 | 21.68 | 15.85 | 9.78 | 58.04 | 14.16 | 31.82 |
> | HyProMeta | 90.81 | 54.35 | 56.34 | 74.83 | 67.92 | 63.33 | 23.78 | 24.39 | 16.22 | 72.73 | 34.20 | 36.48 |
> | **Ours** | **95.80** | **65.56** | **81.53** | **80.42** | **65.94** | **66.99** | **29.37** | **30.31** | **21.11** | **76.92** | **44.79** | **47.77** |
>
>
> >The manuscript requires further refinement in details to improve readability. For instance, "DirectFM" in Table 7 is not defined.
>
> Thank you very much for your comment! DirectFM indicates in this ablation we do not learn residuals but using flow matching to directly generate images. The corresponding description has been added in our revised paper.

---

### Official Review · Reviewer_UAWs · 2025-11-03

**Soundness:** 3
**Presentation:** 3
**Contribution:** 2
**Rating:** 6
**Confidence:** 4

**Summary:**

This paper proposes a novel reliability-aware meta-learning framework called EReLiFM,  that combines evidential loss–based clean/noisy data separation with domain- and category-conditioned residual flow matching. It aims to improve open-set domain generalization under noisy labels by filtering clean samples, expanding them through structured residual flows, and recycling noisy data with evidential pseudo-labeling. Experiments on DG benchmarks show that it achieves SOTA performance, outperforming prior methods across multiple noise settings.

**Strengths:**

1. The paper introduces a novel integration of evidential uncertainty modelling and residual flow matching, effectively addressing noisy labels for open-set domain generalization task.

2. It has proposes a novel framework, EReLiFM that demonstrates strong and consistent performance improvements across multiple benchmarks, showing robustness to different noise levels and backbone architectures.

**Weaknesses:**

1. See questions
2. Below papers are needed to be cited.

[1] Towards Multimodal Open-Set Domain Generalization and Adaptation through Self-supervision, ECCV 2024
[2] OSLoPrompt: Bridging Low-Supervision Challenges and Open-Set Domain Generalization in CLIP, CVPR 2025

**Questions:**

(1) Why are the evidential loss trajectories preferred over the feature-space embeddings for clean/noisy sample separation?

(2) What is the reason behind separating clean data for meta-train and noisy data for meta-test rather than mixing them?

(3) Is it possible to check the sensitivity of separating clean and noisy samples with other unsupervised clustering methods like K-means and DBSCAN ?

---

> ### Author Response · Authors · 2025-11-24
> **Author Response**
>
> > Below papers are needed to be cited.
>
> Thank you for your comments! We have discussed these papers in our related work section in detail.
>
> >Why are the evidential loss trajectories preferred over the feature-space embeddings for clean/noisy sample separation?
>
> Thank you very much for your comment.
>
> First, we clarify the motivation of using evidential loss trajectories. We leverage evidential loss trajectories to incorporate uncertainty awareness into the recorded training dynamics, enabling UTS-ELC to better identify outlier behaviors that subsequently improve the later label noise separation process.
>
> Second, we clarify why evidential loss trajectories are used instead of features. Feature-space embeddings can be highly distorted by noisy labels, causing clean and noisy samples to overlap and reducing clustering reliability (as seen in HyProMeta). Evidential loss trajectories reflect optimization dynamics, which better separate label noisy/clean samples as shown in Table 7 of our revised paper (the related experimental comparison can be found in Table R1-A of this response).
> This leads to a cleaner partition of reliable data, enabling more stable downstream meta-learning and residual-flow modeling.
>
>
> **Table R1-A Comparison with feature based separation**
> | Method            | 20% sym | 50% sym | 80% sym | 50% asym |
> |-------------------|---------|---------|---------|----------|
> | Representation    | 54.80   | 49.74   | 45.32   | 25.07    |
> | **Ours**          | **90.04** | **92.25** | **56.05** | **52.91** |
>
>
> >What is the reason behind separating clean data for meta-train and noisy data for meta-test rather than mixing them?
>
>
> Thank you very much for your comment. Clean data are used in the meta-train stage because the model needs reliable supervision to learn stable and generalizable update directions. If more noisy samples are used during the meta-training stage, their incorrect labels would distort the gradients during meta training and undermine the foundation of the meta-learner, as we need to use the weight update during the meta training to guide the meta test stage.
> Once this more reliable meta training stage is established, the meta-test stage introduces the noisy data, allowing the model to handle them with evidential pseudo-labeling and uncertainty-aware corrections. This clean-then-noisy separation ensures that the model first learns ''how to learn'' from more reliable training samples before being asked to cope with samples with incorrect labels, leading to better optimization.
>
>
> > Is it possible to check the sensitivity of separating clean and noisy samples with other unsupervised clustering methods like K-means and DBSCAN ?
>
> Thank you for your suggestion! We have added the experiments for DBSCAN and K-means in Table 19 of our rebuttal (corresponding to Table R1-B in this response).
> From the experimental results we can observe that DBSCAN and K-means remain highly sensitive to density assumptions and centroid initialization, which often leads to unstable cluster boundaries when loss patterns vary across domains and categories. In contrast, FINCH produces data-driven hierarchical partitions that do not require predefined density thresholds or cluster numbers, allowing it to better adapt to the heterogeneous and noisy loss dynamics characteristic of OSDG-NL. The subsequent GMM refinement further models the aggregated loss statistics with a probabilistic mixture, yielding a smoother and more discriminative clean/noisy separation than the label noisy/clean partitions produced by DBSCAN and K-means. Consequently, our FINCH+GMM pipeline delivers a more reliable clean partition, which directly strengthens the following procedures in our EreLiFM process and leads to superior OSDG-NL performance.
>
> **Table R1-B Comparison with more unsupervised clustering methods, e.g., K-means and DBSCAN.**
> | Method            | 20% sym | 50% sym | 80% sym | 50% asym |
> |-------------------|---------|---------|---------|----------|
> | GMM               | 72.11   | 87.83   | 54.56   | 50.49    |
> | FINCH             | **90.39**   | 85.29   | 50.02   | 38.71    |
> | DBSCAN            | 82.42   | 54.45   | 20.12   | 24.71    |
> | KMEANS            | 80.05   | 50.08   | 20.12   | 24.71    |
> | **Ours**          | 90.04 | **92.25** | **56.05** | **52.91** |

---

### Author Response · Authors · 2025-11-24
**Global Response**

Dear Reviewers,

Thank you very much for your thoughtful and constructive feedback. We sincerely appreciate the time and effort you invested in reviewing our submission.

We have carefully addressed all comments in our individual responses and have uploaded a revised version of the manuscript reflecting the requested changes (highlighted in blue color). We may still make minor refinements as we continue proofreading to ensure clarity and completeness.

Thank you again for helping us improve our work.

Best regards,

The authors.

---

### Comment · Area_Chair_3kNa · 2025-11-27
**Reminder: Engage in Discussions and Finalize Your Rating**

Dear Reviewers,

Thank you for your valuable reviews. With the Reviewer-Author Discussions deadline approaching, please take a moment to read the authors’ rebuttal and the other reviewers’ feedback, and participate in the discussions and respond to the authors. Finally, be sure to complete the “Final Justification” text box and update your “Rating” as needed. Your contribution is greatly appreciated. I will flag irresponsible (final) reviews and/or any reviewers not participating in discussions.

Reviewers are expected to stay engaged in discussions, initiate them, respond to authors’ rebuttal, ask questions, and listen to answers to help clarify remaining issues.

It is not OK to stay quiet.

It is not OK to leave discussions till the last moment.

If authors have resolved your (rebuttal) questions, do tell them so.

If authors have not resolved your (rebuttal) questions, do tell them so too.

Thanks,

AC

---

### Author Response · Authors · 2025-11-29
**Summarization for AC**

Dear AC,

Thank you very much for the time and effort dedicated to reviewing our paper and rebuttal. Below, we provide a concise summary of our paper’s contributions and the key updates made during rebuttal.

---
## **Contribution Summary**

We propose **EReLiFM**, a reliability-aware meta-learning framework for **open-set domain generalization under noisy labels**:

1. **UTS-ELC**: We use evidential loss trajectories for domain- and category-aware clean/noisy separation, providing substantially more reliable partitions than feature- or prototype-based methods.
2. **DC-CRFM** We introduce *domain- and category-conditioned residual flow matching*, the **first method to model domain and categorical transfer using residual flows**, enabling structured and diverse augmentations beyond interpolation-based approaches.
3. **Reliability-aware Meta-Learning**: We integrate the above components into a clean–augment–denoise pipeline that decouples reliable and unreliable supervision.

Across PACS, DigitsDG, TerraInc, and additionally OfficeHome and VLCS (added in rebuttal), **EReLiFM consistently achieves state-of-the-art performance**, outperforming HyProMeta and all other baselines under both symmetric and asymmetric noise.

---

## **Summary of Rebuttal**

### **Reviewer UAWs**
- We clarified the **motivation of evidential loss trajectories** over features for clean/noisy separation in the rebuttal and added new comparison between using loss and feature in **Table R1-A** (**Table 19** in revision).
- We provided a clearer justification for using clean data in meta-train and noisy data in meta-test in the rebuttal, explaining how this design stabilizes meta-learning under label noise.
- We evaluated the sensitivity to mentioned clustering methods (K-Means, DBSCAN) and showed the superiority of our design in **Table R1-B** (**Table 19** in revision).

### **Reviewer HVGE**
- We provided a detailed explanation of UTS-ELC in **Sec. 3.2**, and added **t-SNE** in **Figure 3(a,b)** (Appendix Sec. F), showing our label-noise detection accuracy (92.25% vs. 72.46% for HyProMeta).
- We added **categorical and domain residual visualizations** of DC-CRFM in **Figure 4** (Appendix Sec. J), highlighting smoother and more diverse transitions than MixStyle/FACT.
- We expanded experiments to **OfficeHome** and **VLCS**, with results in **Table 17** and **Table 18** (Appendix Sec. H), showing consistent improvements.
- We clarified the definition of DirectFM in **Sec. 4.8** and updated the description accordingly.

### **Reviewer WyRZ**
- We reorganized the method section with a **new high-level framework** shown in **Figure 1** and introduced a clearer top-down method explanation in **Sec. 3.2–3.4**.
- We highlighted the **inner/outer loop objectives** and added explicit notation and loss weighting in **Alg. 1** (Sec. 3.2).
- We added a **Preliminaries Section K** (Appendix) covering background on FINCH, GMM, and flow matching.
- We added t-SNE (**Figure 3(a,b)**) and **residual visualizations** (**Figure 4**).
- We corrected acronym usage.

### **Reviewer FcXz**
- We clarified the conceptual justification for **clean-as-meta-train and noisy-as-meta-test** in **Sec. 3.2**.
- We confirmed that we already compare against diverse **meta-learning baselines** (MLDG, MEDIC, EBiL-HaDS, HyProMeta), reported in **Tables 1–6** (Main Text).
- We added the behavior under nearly-clean data in **Sec. O and Figure 5 (Appendix)**, showing UTS-ELC naturally shrinks the noisy set to near zero.
- We improved qualitative comparisons between our method and interpolation-based methods in **Sec. J** (Appendix) with supporting visuals in **Figure 4**.
- We clarified ablation cases “w/o UTS-ELC in RFM" and “w/ UTS-LC in RFM" in **Sec. 4.8**.

### **Reviewer Xk5Q**
- We added explanation for **why evidential loss trajectories better separate clean/noisy samples** in **Sec. 3.2** (main text) and rebuttal.
- We illustrated that DiT is only used during inference. We showed via comparisons in **Table 7** and **Tables 1–6** that simpler augmentations (MixStyle, MixUp, ODGNet(GAN)) cannot advances DC-CRFM.
- We added error propogation analysis of UTS-ELC in **Appendix Sec. L**, explaining the difference between noisy→clean vs. clean→noisy mis-assignments.
- More benchmarks are added (OfficeHome & VLCS) with results in **Table 17** and **Table 18**.
- We added hyperparameter sensitivity results and analysis (epoch window \(N_e\)) in **Appendix Sec. F**, supported by **Figure 3(c)**.
- We included statistical significance tests (mean ± std) in **Table 21 (Appendix Sec. M)**.
- We analyzed the pseudo-label accuracy vs. OSCR correlation in **Table 22 in Appendix (Sec. N)**.

### **Overall**
We have significantly improved the paper’s clarity, theoretical grounding, empirical evaluation, and visual evidence. We believe these revisions fully address the reviewers’ and AC’s concerns and substantially strengthen the manuscript.

---

### Meta-Review · Area_Chair_jXZG · 2025-12-30

**Summary:**

This paper addresses a problem named open-set domain generalization under noisy labels (OSDG-NL), which aims to classify unseen categories in new domains with noisy labels. This paper proposes a three-stage framework, named EReLiFM: (1) distinguishing clean/noisy labels via UTS-ELC (Unsupervised Two-Stage Evidential Loss Clustering), (2) augmenting training data with DC-CRFM (Domain- and Category-Conditioned Residual Flow Matching), and (3) meta-learning-based optimization. Experiments are conducted on PACS, DigitsDG, and TerraINC datasets.

**Reviewer Concerns:**

This paper has four borderline initial recommendations (mostly negative). The main concerns raised by the reviewers can be summarized as follows:

- Lack of justification of UTS-ELC (UAWs, HVGE, Xk5Q)
    - The rebuttal comment clarifies that a feature-based separation is potentially sensitive to outliers, but it does not justify the working mechanism of the proposed UTS-ELC. More specifically, the argument "the utilization of the training dynamics is more reliable under domain shift and label noise" should be supported more rigorously with a theoretical or conceptual explanation.
- Lack of justification for separating clean/noisy for meta-train and meta-test (UAWs, FcXz)
    - The rebuttal comment clarifies the goal of meta-train and meta-test. However, this heavily relies on plain text explanation without a concrete and rigorous description of the given context. Again, this should be supported by a theoretical or conceptual explanation. The current explanation cannot answer the question "why do we need this approach to solve the problem?"
- Lack of justification/visualization of DC-CRFM (HVGE, WyRZ, FcXz, Xk5Q)
    - The revision includes visualization results and additional comparison results with a GAN-based augmentation. However, Mixup and ODGNet are not the only augmentation methods. There is no rationale behind the selection of the comparison methods. For example, there are many augmentation methods, including instance-wise augmentations (e.g., randaug, cutout), mix-based augmentations (e.g., cutmix, manifold mixup), and generative model-based augmentations. There is no specific reason why Mixup is the only comparison method.
- Not enough benchmark, hyperparameter sensitivity studies, design choice studies, and comparison methods (HVGE, FcXz, Xk5Q)
    - As stated by the DomainBed paper, domain generalization is very sensitive to the hyperparameter choice, design choice, and benchmarks. Therefore, as many reviewers pointed out, I think that it is very important to compare the proposed method in a comprehensive experimental setting. The rebuttal document includes a few additional results, such as OfficeHome and VLCS benchmarks. However, as some reviewers mentioned, it is important to examine the method on larger datasets with a large number of labels (e.g., DomainNet) and significant domain shift (e.g., bio dataset). The current experiments look insufficient to satisfy this. Furthermore, the hyperparameter selection should follow the rigorous evaluation protocol (e.g., using a held-out validation split, rather than directly tuning the hyperparameters on the test split) as pointed out by the DomainBed paper. I think that the experimental results can be enhanced to improve the submission.
- Weak motivation for studying open set domain generalization and noisy labels simultaneously (FcXz)
    - The rebuttal comment mentions that the combination of the problems has been largely overlooked. In my opinion, it does not answer the question. It only describes that there are not many previous works on this problem, not the importance of the problem.

There are some concerns addressed by the rebuttal comment.

- Lack of sensitivity analysis (or failure mode analysis) of clean/noisy separation (UAWs, Xk5Q)
    - The rebuttal comment includes an additional experiment result with DBSCAN/K-means clustering. Also, the revised paper includes the robustness experiment for the failure mode analysis.
- The proposed method will not work well under 100% clean data (FcXz)
    - The rebuttal comment clarifies the working mechanism of the proposed method. If everything works perfectly, then the proposed method will work well.

There are minor concerns, such as presentation quality (HVGE, WyRZ), computational overhead (Xk5Q), cross-dataset generalization (Xk5Q), and statistical significance (Xk5Q).

In summary, as the reviewers pointed out, the proposed method contains a lot of complex design choices that are not well justified. The rebuttal comment and the revised paper partially addressed the concerns, but the AC thinks that the current version still cannot answer the question "why do we need this approach to solve the problem?". Similarly, I think that the main motivation of the proposed problem is also somewhat weak (as well as the proposed design choices). I think that the submission will be stronger if the manuscript is improved. Furthermore, the evaluation protocol and benchmarks look somewhat insufficient, especially considering the noisy domain generalization results (as shown by the DomainBed paper). Overall, I recommend rejecting this paper.

**Reviewer Scores:**

I don't think each review went over the borderline, even if there were active discussions between the authors and the reviewers. The main concerns raised by the reviewers were mostly about the justification of the proposed method and modules. They are somewhat difficult to address in the rebuttal comment without heavy major revision.

---

### Decision · Program_Chairs · 2026-01-26

Reject